Learning new word meanings from story reading: the benefit of immediate testing

http://orcid.org/0000-0002-9596-7729 Hulme Rachael C. 1 2 rachael.hulme.14@ucl.ac.uk
http://orcid.org/0000-0002-8608-7244 Rodd Jennifer M. 1
1 Department of Experimental Psychology, University College London , London , UK
2 Aston Institute of Health and Neurodevelopment and School of Psychology, Aston University , Birmingham , UK
Barnhart Anthony
Electronic publication date: 2021 Aug 10
Publication date: 2021
Volume: 9
Electronic Location ID: e11693
Received 2021 Mar 11; Accepted 2021 Jun 8
Copyright: © 2021 Hulme and Rodd
Copyright year: 2021
Copyright holder: Hulme and Rodd
License: This is an open access article distributed under the terms of the Creative Commons Attribution License, which permits unrestricted use, distribution, reproduction and adaptation in any medium and for any purpose provided that it is properly attributed. For attribution, the original author(s), title, publication source (PeerJ) and either DOI or URL of the article must be cited.
License URL: https://creativecommons.org/licenses/by/4.0/

Keywords: Incidental learning, Intentional learning, Testing effect, Homonyms, Story reading

Funding: Doctoral Studentship 1473923 Economic and Social Research Council ES/K013351/1 This work was supported by a doctoral studentship (award ref 1473923) from the Economic and Social Research Council (grant number ES/K013351/1). There was no additional external funding received for this study. The funders had no role in study design, data collection and analysis, decision to publish, or preparation of the manuscript.

==============================
This study investigated how word meanings can be learned from natural story reading. Three experiments with adult participants compared naturalistic incidental learning with intentional learning of new meanings for familiar words, and examined the role of immediate tests in maintaining memory of new word meanings. In Experiment 1, participants learned new meanings for familiar words through incidental (story reading) and intentional (definition training task) conditions. Memory was tested with cued recall of meanings and multiple-choice meaning-to-word matching immediately and 24 h later. Results for both measures showed higher accuracy for intentional learning, which was also more time efficient than incidental learning. However, there was reasonably good learning from both methods, and items learned incidentally through stories appeared less susceptible to forgetting over 24 h. It was possible that retrieval practice at the immediate test may have aided learning and improved memory of new word meanings 24 h later, especially for the incidental story reading condition. Two preregistered experiments then examined the role of immediate testing in long-term retention of new meanings for familiar words. There was a strong testing effect for word meanings learned through intentional and incidental conditions (Experiment 2), which was non-significantly larger for items learned incidentally through stories. Both cued recall and multiple-choice tests were each individually sufficient to enhance retention compared to having no immediate test (Experiment 3), with a larger learning boost from multiple-choice. This research emphasises (i) the resilience of word meanings learned incidentally through stories and (ii) the key role that testing can play in boosting vocabulary learning from story reading.

Introduction

Recent research emphasises the importance of good vocabulary knowledge: individuals with better vocabulary perform better on reading comprehension tests, and have better educational outcomes (Armstrong et al., 2017; Cain & Oakhill, 2014). Despite widespread acceptance that incidental learning from natural linguistic environments (e.g., conversations, books, TV) is the main source of vocabulary learning (Batterink & Neville, 2011; Nagy, Herman & Anderson, 1985; Nagy, Anderson & Herman, 1987), most studies of vocabulary learning in adults use highly artificial stimuli, tasks, and learning conditions. The current experiments focus on learning from naturalistic fiction stories that are read by participants in their native language without any explicit instructions to learn the new vocabulary that the stories contain. We investigate the extent to which people’s ability to retain newly-learned word meanings over time is improved by requiring them to retrieve these word meanings during the intervening period between encoding and a later test. The presence of such a beneficial “testing effect” has been well established through studies of explicit, intentional learning (for reviews, see Roediger & Butler, 2011; Rowland, 2014), but it is unclear whether retrieval would similarly enhance memory for vocabulary learned under more naturalistic, incidental learning conditions. The finding that vocabulary learning from naturalistic materials could be significantly boosted by a brief episode of testing could provide a simple approach to boosting vocabulary gains in real-world settings.

Incidental vocabulary learning is defined as learning words and their meanings whilst engaged in another activity such as listening or reading for comprehension (Hulstijn, 2003). A real-life context in which adults often learn new words and their meanings is when reading fiction, due to the rich and varied situations that are often depicted (Nation, 2017). Studies of word learning from stories by adult native-language (L1) readers have adopted highly naturalistic methods by using either authentic texts (Godfroid et al., 2017; Saragi, Nation & Meister, 1978) or texts modified or written specifically for the purposes of the studies (Batterink & Neville, 2011; Henderson et al., 2015; Pellicer-Sánchez, 2016). In these studies participants read works of fiction with the primary focus being on comprehension, with vocabulary learning as a by-product. To discourage intentional learning strategies, readers are not given any instruction to learn new vocabulary encountered in a text and are not informed that their memory will later be tested.

The current study uses a paradigm developed by Hulme, Barsky & Rodd (2019) in which participants encounter artificial new meanings for familiar English words in the context of custom-written short stories (e.g., learning that a foam is a type of safe concealed within a piece of furniture). This ability to learn new word meanings is a key aspect of vocabulary development: around 80% of common English words have more than one definition (Rodd, 2018; Rodd, Gaskell & Marslen-Wilson, 2002). Adults often learn additional word senses/meanings, and continue to update their knowledge of these words throughout their adult lives (Betts et al., 2018; Gaskell, Cairney & Rodd, 2019; Gilbert et al., 2018, 2021; Rodd et al., 2013). Examples of reasons why adults learn new meanings for familiar words include language evolution (e.g., the internet-related meaning of “troll”), or learning a new subject or activity (e.g., the sailing term “boom”; Eligio & Kaschak, 2021; Rodd et al., 2012, 2016). New meanings are often learned when reading stories, especially of the science fiction or fantasy genres (e.g., a “galleon” is a coin of the wizarding currency in the Harry Potter series of novels by J. K. Rowling). Recently, Fang, Perfetti & Stafura (2016) proposed that learning new meanings for familiar words is a dual-phase process whereby familiarity with the word form may facilitate learning with the initial encounters, but inhibition due to meaning competition begins to take effect after subsequent exposures to the newly ambiguous word (Maciejewski et al., 2020; Maciejewski & Klepousniotou, 2020; see Rodd, 2020 for review).

Hulme, Barsky & Rodd (2019) found that participants were able to recall the new meanings for the known words reasonably well (38.5% correct) after only two exposures in a story context, with a linear increase in meaning recall with additional exposures (63.5% correct after eight exposures). Interestingly, Hulme, Barsky & Rodd (2019)’s participants showed no significant forgetting of the new meanings they had learned at a surprise test one week later across all of the exposure conditions. The current study further examines this incidental form of word learning by (i) comparing performance to a more explicit learning condition and (ii) investigating the potential boost to performance from an immediate test of knowledge after training. Understanding how these two factors impact on long-term retention of vocabulary will provide a critical foundation for subsequent development of interventions to boost vocabulary acquisition.

The conditions of initial vocabulary acquisition (incidental or intentional) prompt different types of information processing, which may affect retention of word meanings in different ways. Vocabulary learned under intentional conditions may be retained better over time because more attention is directly focussed on encoding the word meanings, and the meaning is made more explicit. This more strategic processing might be particularly important for facilitating access to prior knowledge in the case of learning new ambiguous words where the learner may benefit from more explicitly noticing the mismatch between the familiar word meaning and the new meaning. In contrast, incidental vocabulary learning from story reading may benefit from the rich and informative story contexts (Webb, 2008), and it has been suggested that the increased mental effort required to encode new word meanings inferred from context may be beneficial for retention (Hulstijn, 1992). However, it is also important to consider that while incidental vocabulary learning is usually contextualised (with words embedded in informative contexts from which meaning is inferred), intentional vocabulary learning may also involve context, or it can be decontextualised.

The consensus from the literature on adult second language (L2) learning (e.g., Hulstijn, 1992; Lehmann, 2007; Peters et al., 2009), and research with teenagers learning L1 vocabulary (Konopak et al., 1987) is that intentional learning offers greater vocabulary gains and is more efficient than incidental learning. However, some other studies have found little difference (Lehmann, 2007), or even an efficiency advantage in terms of words learned per min for incidental learning (Mason & Krashen, 2004). Several recent studies with adult L1 readers have also found good levels of native language vocabulary acquisition from reading alone (Batterink & Neville, 2011; Godfroid et al., 2017; Pellicer-Sánchez, 2016). A further key factor that could differ between vocabulary acquisition under incidental and intentional learning conditions is the impact of testing on subsequent retention.

The “testing effect” refers to the finding that testing memory following training can enhance long-term retention, as the additional retrieval practice at test affords an opportunity for further learning (for reviews, see Roediger & Butler, 2011; Rowland, 2014). The effect has been demonstrated as robust in various experiments using explicit, intentional learning conditions. However, it is unclear whether the testing effect would provide a similar benefit for vocabulary learned under incidental learning conditions. Given that the vast majority of native language words and their meanings are learned incidentally (Batterink & Neville, 2011), it is important to examine the impact of the testing effect under such learning conditions. If the presence of a quick, immediate vocabulary test can indeed enhance learning/retention for incidentally learned vocabulary this could potentially provide a simple method for boosting vocabulary gains from story reading, especially within educational settings.

In vocabulary learning research, retrieval practice has been shown to lead to better retention of new words over time with adults learning second language (L2) vocabulary under intentional conditions (e.g., Fritz et al., 2007; Karpicke & Roediger, 2008; Van den Broek et al., 2013, 2018), and similarly with children learning novel L1 words (Goossens et al., 2014a, 2014b; Toppino & Cohen, 2009). The testing effect further enhances retention when feedback is provided on performance on the immediate test (e.g., Pashler et al., 2005), but retrieval practice is often beneficial even in the absence of any feedback (Roediger & Butler, 2011). The precise neurocognitive mechanism underlying the testing effect is currently unclear, but it has recently been suggested that retrieval practice may provide a fast track to consolidation of new information through the online reactivation of related knowledge (Antony et al., 2017; see the General Discussion for further discussion).

It is possible that different learning conditions preceding retrieval practice could moderate the testing effect for various reasons. For example, it is thought that semantic elaboration may be key to the neurocognitive mechanism underlying the testing effect (Carpenter, 2009). If this is the case, then the richer story contexts during encoding in the incidental condition could provide more fertile material for semantic elaboration, thus enhancing the testing effect. On the other hand, research has suggested that the benefits of retrieval practice are greater when retrieval success during practice is high (Rowland, 2014). Therefore, if intentional learning is more effective than incidental learning then this could lead to a stronger testing effect following encoding under intentional learning conditions.

Retrieval practice has been shown to benefit long-term retention of information learned under a variety of conditions (e.g., Butler, 2010; Karpicke & Roediger, 2008; Roediger & Karpicke, 2006a; Van den Broek et al., 2013), although little research has compared across different learning conditions. One study (Goossens et al., 2014a) directly compared the impact of testing on children’s learning of novel L1 vocabulary from a story context to learning new words in isolation. Results showed that children correctly recalled more word meanings that had been tested, and children in the word list condition remembered the word meanings better overall than those in the story condition. The testing effect was also slightly stronger for the word list condition. However, learning was not incidental in either condition in this study, and children who heard the story also had the meanings of the words explained to them. Furthermore, the participants in this study were children (aged 8–11), and results may differ for adults whose advanced language skills and vocabulary knowledge make them better equipped to learn more successfully from the richer contexts that stories provide. It therefore remains to be seen whether the benefit of retrieval practice would differ for the learning of new word meanings acquired solely under incidental conditions in a story context, as compared with learning under intentional conditions.

Experiment 1: incidental versus intentional learning

Experiment 1 compared the story-reading method designed by Hulme, Barsky & Rodd (2019) for studying incidental learning of new meanings for familiar words with a more conventional, intentional training procedure. This provided a baseline assessment of how well adults are able to learn new word meanings from a naturalistic incidental learning paradigm as compared to a more conventional explicit approach to vocabulary learning, and provided a foundation for the subsequent preregistered experiments to investigate learning performance in more detail. Specifically, Experiments 2 and 3 follow up on Experiment 1 to examine whether the inclusion of an immediate test of new vocabulary knowledge aids learning and improves memory of new word meanings 24 h later. This may be especially pertinent for vocabulary acquired through incidental learning conditions as it may prompt participants to adopt different information processing strategies after initial acquisition.

In Experiment 1 participants learned novel meanings for existing unambiguous words through both incidental story-reading (as in the study by Hulme, Barsky & Rodd, 2019) and a newly developed intentional task-based learning procedure, with the same number of exposures to items. The two learning conditions were implemented based on typical paradigms for these two types of learning. However, it is important to note that there are multiple differences in the learning experience, for example only the incidental learning paradigm required participants to infer meaning from context. While it is more common for incidental learning to be contextualised in this way, some intentional learning paradigms also involve contextualised learning (see for example: Van den Broek et al., 2018). The stories used in the incidental learning condition combined naturalistic elements of authentic texts (Godfroid et al., 2017; Saragi, Nation & Meister, 1978) with precise experimental control over the exposure to items within the text (Batterink & Neville, 2011; Pellicer-Sánchez, 2016). Items were encountered incidentally within the stories that participants read for comprehension and were central to the narrative.

Participants’ knowledge of the new meanings for all items was assessed first through cued recall, and second through a multiple-choice meaning-to-word matching test. The recall measure is a harder test with fewer cues to help retrieve memories of the new word meanings, while the multiple-choice test is a recognition measure with more cues and is therefore the easier of the two tests. Using two tests of learning with different difficulty levels allowed us to reduce the possibility of floor/ceiling effects. The tests were administered both immediately after learning, and again 24 h later to assess longer-term retention. Based on the previous research, we predicted that learning of new word meanings would be better for the intentional learning condition, although we expected reasonably good vocabulary learning for the incidental learning condition in line with the findings of Hulme, Barsky & Rodd (2019). We also predicted there would be little forgetting after 24 h, although we had no specific predictions as to whether this would differ for new word meanings acquired through incidental or intentional learning conditions. Our predictions were the same for the cued recall test the and multiple-choice meaning-to-word matching test.

The materials, data, and analysis scripts for Experiment 1 can be found on the Open Science Framework (OSF; https://osf.io/k32tw). For all experiments we report all measures, conditions, data exclusions, and how we established the sample size.

Method

Participants

We aimed to recruit 40 participants for Experiment 1. The study by Hulme, Barsky & Rodd (2019) included 64 participants who were trained on four items (one per exposure condition) in one of 16 experiment versions (four participants per version). In this study participants were trained on eight items (four items per learning condition) in one of eight experiment versions (five participants per version), we therefore expected power to be comparable to that of Hulme, Barsky & Rodd (2019). Forty participants were included in the experiment (age: M = 30.1 years, SD = 7.1; 23 female). Participants were recruited through the Prolific recruitment website (Damer & Bradley, 2014) using pre-screening criteria. They gave their informed consent before taking part (by means of ticking boxes in the online consent form). The UCL Experimental Psychology Ethics Committee granted ethical approval for the research (Ref: EP/2017/009). Participants were invited to take part if they were a current UK resident, a monolingual native speaker of British English, and had no diagnosis of reading or language impairments. They were paid for their participation in the first session of the experiment (£5) and additionally upon completion of the second session 24 h later (£1). Of the 40 participants who completed the first session, 31 also completed the 24-h follow-up session on time (77.5%). One additional participant was excluded from the second session due to completing it after the deadline (within 6 h of receiving the invitation for the follow-up session).

Five additional participants were excluded from the study—two were not monolingual native British English speakers, and three got more than one of the multiple-choice comprehension questions wrong when reading the stories (see Procedure). Excluded participants were replaced during recruitment.

Materials

Novel word meanings

The stimuli were 16 real English nouns that were given artificial new meanings, taken from the study by Hulme, Barsky & Rodd (2019) (see Table S1 for the stimuli: https://osf.io/m4wxa). The new meanings were unrelated to the existing meanings of the words, and described hypothetical innovations, discoveries, and inventions. There was one definition sentence for each of the stimulus words that described its new meaning, for example: “A foam is a safe that is incorporated into a piece of furniture with a wooden panel concealing the key lock, and each is individually handcrafted so that no intruders are able to recognize the chief use of the furniture.” The sentences were matched for length (M = 32.9 words, SD = 3.7). Each new meaning had three distinguishing semantic features to maintain a similar level of complexity for each new concept, for example, for foam: “a safe inside a piece of furniture,” “has a hidden key lock,” and “individually handcrafted to fool intruders.” The words and their meanings were incorporated into story narratives for the incidental learning condition, and the definition sentences were presented to participants in the definition reading phase of the intentional learning condition.

Three shorter paraphrased excerpts of the definition sentences were created for use in the two-alternative multiple-choice training task (length: M = 11.29 words; SD = 2.13), with each sentence describing a semantic feature of that item (e.g., for “foam”: “A secure place to store valuables within an item of furniture.”; “A safe with a wooden panel disguising the key lock.”; and “A bespoke handcrafted piece of furniture containing a safe hidden from intruders.”). Paraphrased versions were used to encourage participants to read the whole sentence each time, rather than relying on recognition of the first words (see Table S2 for the short sentences: https://osf.io/m4wxa).

An additional longer paraphrased version of each of the definition sentences (which were used in the test of cued recall of word forms in Hulme, Barsky & Rodd (2019)’s study) were used in the multiple-choice test at the end of this experiment (see Table S3 for the sentences used for the multiple-choice test: https://osf.io/m4wxa).

Short stories

The four short stories from Hulme, Barsky & Rodd (2019)’s study were used to present stimuli to participants in the incidental learning condition in this experiment (see the Supplementary Materials for the stories: https://osf.io/m4wxa). These stories (ranging 2307-2446 words in length) were written by a professional children’s author (Story 1: Pink Candy Dream), and an unpublished author (Story 2: Prisons, Story 3: Reflections upon a Tribe, and Story 4: The Island and Elsewhere), and were designed to be interesting for an adult audience. Each story incorporated four of the items in the context of their new meanings, with each item appearing a total of eight times at naturally distributed positions within a story. No item appeared in more than four consecutive sentences, and all items occurred on at least two different pages of the story. On the first presentation of a stimulus word, sufficient information was given to allow the reader to derive the new meaning from the context from the first exposure, for example, “‘Yes,’ I murmured, breathing again. ‘I knew it! It’s a foam.’ The ornate chaise longue was no ordinary piece of furniture but concealed a built-in safe with an intricate key-operated locking system.” The amount of information about each new meaning in subsequent exposures varied naturally with the story narratives. A degree of inference was required to extract the meaning from the context to reflect natural word learning from reading where explicit definitions are rarely given.

Design

The experiment employed a within-participants and within-items design: participants were trained on four items through the incidental learning condition and four items through the intentional learning condition. Each participant was trained on only half the total number of stimuli as this was deemed to be a feasible number of new meanings to learn in a single session. To ensure each new word meaning was seen an even number of times in each condition, and that the order of learning conditions was counterbalanced across participants (to minimise any order effects), we created eight versions of the experiment. Participants were pseudorandomly assigned to one of the eight versions of the experiment. Time of test (immediate vs. 24 h later) was also within-participants (based on the 31 participants who completed both sessions). The dependent measures were accuracy in cued recall of the new word meanings, and accuracy in the multiple-choice test.

Procedure

The experiment was run online using Qualtrics (Qualtrics, 2015), and participants were instructed to complete each session in one sitting without breaks. Participants were asked to read the stories and definitions carefully, and were not told that their memory for the new word meanings would be tested. Participants were told that the aim of the experiment was to investigate subjective reading style and comprehension. After completing the first session, participants were also not informed that they would be contacted at the same time the following day to invite them to complete the second session to discourage the use of deliberate memorisation strategies.

Participants read one of the short stories in the incidental learning condition. Each story was divided into five pages of roughly even length and displayed on-screen one page at a time. After each page, a multiple-choice comprehension question appeared on a separate screen asking about details of the story’s plot from the preceding page (without probing details of the novel word meanings). Participants were instructed to read the story closely, and to answer a multiple-choice comprehension question after each page. Participants were able to re-read sentences on the current page, although no instructions were given to participants on this; they were not able to go back to reread previous pages. For the comprehension questions participants had to select the correct answer from four options that appeared in a randomised order; they were designed to be very easy for any participant who had attentively read the text. Participants were excluded if they got more than one of the comprehension questions wrong.

The intentional learning condition consisted of two phases which both repeated once: definition sentence reading, followed by two-alternative multiple-choice meaning-to-word matching. In the definition reading phase, participants were presented with sentences that described the key semantic features of each of the novel word meanings, stating the word to which it referred. These four definition sentences were presented one at a time on separate pages, and the order of presentation was randomised for each participant. Participants were instructed to read each definition carefully to make sure they understood it before proceeding to the next one.

Once participants had read all of the definition sentences once, they moved immediately on to the two-alternative multiple-choice meaning-to-word matching task. Participants were presented one at a time with three shortened, paraphrased versions of the definitions of each of the novel meanings. For each item participants were instructed to choose the correct word for the new meaning from two options: the correct word and one foil word. After selecting one of the options, participants were provided with feedback, which either said “Correct answer!” or “Incorrect.”. The short sentences were presented in a pseudorandomised order, ensuring that the sentences referring to each item were roughly evenly spaced, and none referring to the same item occurred one after another. The foil word for each trial was one of the other words from the intentional training condition. Each foil word was paired an even number of times with each correct word, and the order that the correct word and foil appeared in was randomised for each trial. The two phases of the intentional training were then repeated in the same order. This gave a total of two exposures to the novel word meanings from the definition sentence reading phase and six exposures to the new meanings from the multiple-choice task, totalling eight exposures—equal to the number of exposures in the incidental learning condition. Participants spent more time reading the story (including comprehension questions; M = 12 mins 30 s, SD = 4 mins 34 s) than they spent on the intentional training task (M = 5 mins 28 s, SD = 2 mins 55 s), t(39) = 11.43, p < .001.

After they had completed training through both the incidental and intentional learning conditions, participants completed a brief filler task. This was the 34-item version of the Mill Hill vocabulary test (Mill Hill Vocabulary Test, Set A: Multiple Choice: Raven, Raven & Court, 1998). For each item, participants were required to select one word from a list of six options that most closely matched the meaning of the presented word. None of the stimulus items appeared in the filler task. The purpose of this task was to counteract any recency effects of memory for stimulus items encountered toward the end of training; responses were not analysed.

Participants were next given a cued recall test of all eight of the new meanings they had encountered in the experiment. Participants saw each of the eight words they had been trained on and were asked to recall the appropriate new meaning and type it into a text box. They were encouraged to provide as much detail as possible and to try to answer in full sentences even if they were unsure of their answer. If they could not remember anything about the new meaning for the word, they were instructed to type “don’t know.” The order of presentation of the words was randomised for each participant, with the four words from each training method randomly intermixed within the test; this was also the case for the subsequent test.

The second test was an eight-alternative multiple-choice meaning-to-word matching test. Participants were presented one at a time with paraphrased definitions of the novel word meanings they had been trained on. The sentences omitted the words to which they were referring, and for each novel meaning participants were asked to select the word that they thought matched the definition from a list of all eight of the stimulus words they had encountered throughout the experiment. The order of the eight words to choose from was randomised for each test item, and the order of presentation of the new meanings was randomised for each participant.

Finally, participants provided their demographics details and answered some questions about their reading habits. These questions were used to maintain the impression that the experiment was investigating general reading and comprehension and the responses were not analysed.

Exactly 24 h after the first session, participants were invited to take part in a short 24-h follow-up to the experiment. Thirty-one participants completed the follow-up tests, which they did an average of 24 h and 1 min (SD = 54 mins, range = 22 h 26 mins–28 h 2 mins) after the first session. The follow-up tests consisted of a repeat of the two tests from the first session of the experiment in the same order.

Results

Analysis procedure

Responses from the multiple-choice test were either coded as “1” for correct or “0” for incorrect with regards to which word had been selected to match with the meaning. Responses for the cued recall test were independently coded for accuracy by the experimenter and a research assistant, blind to condition, as either “1” for correctly recalled meanings or “0” for incorrect1 . Responses were leniently coded as correct if at least one correct semantic feature was recalled. Any ambiguous or partially correct responses were resolved on a case-by-case basis through discussion. One item (“bruise”) was excluded from the analyses for the cued recall measure, as the percentage of participants who gave a correct response for that item in one of the two learning conditions (incidental, 20.0%) was more than two standard deviations below the grand mean for all items across both learning conditions (M = 72.3%; SD = 23.5).

Data were analysed with logistic mixed effects models using the lme4 package (version 1.1–7; Bates et al., 2015) and R statistical software (version 3.0.2; R Core Team, 2017). Four separate models were created: one for each test measure comparing the accuracy between day one and day two (including only the participants who completed both test sessions, N = 31), and for each measure for all participants tested on day one only (N = 40). These latter analyses aimed to verify that the data from this larger set of participants did not differ from the subset who chose to complete both sessions.

Due to a potential effect of counterbalancing order, in that participants who completed the intentional condition first may suspect that they would be tested on the words in the stories, we included a factor for the order of the learning conditions in our models. The contrasts for the fixed effects were defined using deviation coding for learning condition (incidental: −0.5, intentional: 0.5), and the order of the learning conditions (first: −0.5, second: 0.5), with the interaction coded by multiplying the contrasts for these two factors. The two models comparing performance between day one and day two contained additional fixed effects for time (day one: −0.5, day two: 0.5), and the interactions between time and learning condition, time and learning order, and the three-way interaction. Random effects structures were determined by identifying the maximal model (Barr et al., 2013). This included by-participant and by-item random intercepts, and by-participant and by-item random slopes for learning condition2 . The models comparing performance between day 1 and day 2 also included by-participant and by-item random slopes for time and the interaction between time and learning condition. Where the maximal model failed to converge, we simplified the models by removing the correlations between the by-participant and by-item random slopes and random intercepts without removing any of the random slopes, as recommended by Barr et al. (2013). Significance of the fixed effects and interactions was assessed using likelihood ratio tests comparing the full model to models with each fixed factor/interaction of interest removed in turn (but leaving in any interaction involving a factor of interest that has been removed). Follow-up analyses were carried out in the case of any significant interaction using the same method. The p-values for the simple effects analyses were compared against a Bonferroni-corrected α of .025. All data and analysis scripts for this experiment are available via the Open Science Framework (OSF; https://osf.io/k32tw).

Cued recall of meanings

The accuracy data for cued recall of the new meanings comparing performance between day one and day two (N = 31; see Fig. 1) showed a reasonably high level of accuracy for items learned through both learning conditions. But accuracy was significantly higher overall for items learned through the intentional learning condition (day one: 85.2%; day two: 84.9%) than those learned under incidental conditions (day one: 62.1%; day two: 70.1%) (χ2(1) = 14.32, p < .001). There was no significant main effect of time of test (χ2(1) = 1.23, p = .268), nor of learning order (χ2(1) = 0.83, p = .362). While accuracy improved slightly between day one and day two for items learned through the stories and remained at a similar level for items learned through the intentional condition, the interaction between learning condition and time was non-significant (χ2(1) = 1.57, p = .210). The interactions between learning condition and learning order (χ2(1) = 2.16, p = .141), learning order and time of test (χ2(1) = 1.79, p = .181), and the three-way interaction (χ2(1) = 0.83, p = .361) were also non-significant.

Figure 1 Experiment 1. Mean percentage of correct responses on the cued recall test for each learning condition, when tested on day one (immediately after learning) and 24 h later (N = 31).

The LME analyses were carried out on the raw binary accuracy data, however mean percentage accuracy data are displayed in the graphs. Error bars show standard errors for the means, adjusted for the within-participants design (Cousineau, 2005).

The accuracy data for all participants on day one only (N = 40) showed a similar pattern: significantly higher accuracy for items learned through the intentional training condition than the incidental condition (χ2(1) = 21.35, p < .001), and no significant main effect of learning order (χ2(1) = 0.0007, p = .979). However, there was a significant interaction between learning condition and learning order (χ2(1) = 4.68, p = .030): accuracy was higher for items learned through the stories when this had been the first condition in the experiment (70.0%) than when this had been the second condition (47.5%), whilst the opposite was the case for items learned through the intentional condition (first: 82.1%; second: 95.0%). This unexpected result could have been driven by participants becoming somewhat fatigued by the second task and this fatigue effect having greater impact on the story reading task, which took longer. However, this does not perhaps explain why participants’ performance in the intentional condition was slightly higher when it was the second condition in the experiment rather than the first condition.

Multiple-choice meaning-to-word matching

The results for accuracy on the multiple-choice meaning-to-word matching test comparing results between day one and day two (N = 31) are shown in Fig. 2. Accuracy was high in both learning conditions, but was slightly higher for the intentional learning condition (day one: 96.0%; day two: 87.9%) than for the incidental learning condition (day one: 83.9%; day two: 83.1%), although this difference was non-significant (χ2(1) = 3.66, p = .056). The main effect of time was also non-significant (χ2(1) = 3.81, p = .051), as was the main effect of learning order (χ2(1) = 0.002, p = .966). Interestingly, the interaction between learning condition and time was significant (χ2(1) = 3.85, p = .05). The interaction between learning condition and learning order was non-significant (χ2(1) = 2.24, p = .135), as was the interaction between time and learning order (χ2(1) = 0.01, p = .929), and the three-way interaction (χ2(1) = 0.48, p = .488).

Figure 2 Experiment 1. Mean percentage of correct responses on the multiple-choice test for each learning condition, when tested on day one (immediately after learning) and day two (24 h later; N = 31).

Error bars show standard errors for the means, adjusted for the within-participants factor of learning condition (Cousineau, 2005).

To follow up on the significant interaction between learning condition and time, two simple effects analyses were carried out to determine the significance of time within each of the two learning conditions separately. For the incidental learning condition there was no significant effect of time (χ2(1) = 0.10, p = .750), indicating no forgetting between day one and day two. However, there was a significant effect of time for the intentional learning condition (χ2(1) = 6.07, p = .014), which was slightly lower on day two than day one. (The p-values for these simple effects analyses were compared against a Bonferroni-corrected α of .025).

In the accuracy data for the multiple-choice test for all participants on day one only (N = 40), accuracy was again high overall for both conditions. Accuracy appeared slightly higher for items learned through the intentional condition (95.7%) than for items learned through the incidental condition (81.9%), but there was no significant main effect of learning condition (χ2(1) = 1.18, p = .277). There was also no significant main effect of learning order (χ2(1) = 0.004, p = .950), and no significant interaction (χ2(1) = 2.52, p = .113).

Discussion

Experiment 1 aimed to determine how easily novel meanings for familiar words can be acquired incidentally through story reading, as compared with a more intentional learning procedure. The results showed that accuracy in cued recall of new word meanings was significantly higher in the intentional learning condition than the incidental learning condition: 85.2% compared with 62.1% when measured immediately after training. The accuracy data for the multiple-choice meaning-to-word matching test showed a similar pattern (96.0% for the intentional condition and 83.9% for the incidental condition at the immediate test), although for this measure the main effect of learning condition was non-significant. Furthermore, reading the story took participants a significantly longer amount of time, so more was learned in a shorter amount of time through the intentional training task.

These findings are broadly consistent with those of L2 vocabulary learning studies that have found intentional learning to be more efficient than incidental learning of vocabulary (Hulstijn, 1992; Peters et al., 2009). Although recall accuracy was higher for the intentional learning condition, there was also a reasonably high level of acquisition of new meanings for familiar words through the incidental learning condition. This is very similar to the results of the study by Hulme, Barsky & Rodd (2019), where accuracy in recalling new meanings for familiar words was 63.5% after eight exposures in an incidental learning context (accuracy in cued recall of word forms was 69.2%). These results therefore support the findings of recent studies showing good acquisition of L1 vocabulary from reading (Batterink & Neville, 2011; Godfroid et al., 2017; Pellicer-Sánchez, 2016). An additional consideration is that we included multiple-choice comprehension questions between pages of the stories in our incidental learning condition. While these did not probe details of the new word meanings, it is possible that they could have improved comprehension by increasing metacognitive awareness of the comprehension process, similar to a “guided learning” scenario (Blything, Hardie & Cain, 2020).

Interestingly, the multiple-choice test showed a significant interaction between learning condition and time of test, although there was no such significant interaction in the recall measure. Following up on the significant interaction in the multiple-choice measure, simple effects analyses showed that there was significant forgetting after 24 h of items learned in the intentional condition (8% reduction in accuracy between the immediate test and delayed test), but there was no forgetting of items learned incidentally through story reading. This absence of significant forgetting after a 24-h delay replicates the finding from Hulme, Barsky & Rodd (2019) who reported no significant forgetting on this paradigm after a seven-day delay. A possible explanation for this lack of forgetting is that new word meanings learned in a more semantically rich context, such as from stories, may be retained better. New word meanings encountered in stories contain additional contextual information relating to the narrative (e.g., characters’ thoughts and feelings), providing additional cues for participants to rely on for later retrieval. The stories are also more interesting and engaging so may be more memorable for participants in general. However, another possibility is that this difference is a function of initial learning level, as initial performance was very high in the intentional condition and so had further to fall. A further alternative explanation for this finding is that the lack of forgetting of items learned incidentally through story reading could be due to memory reactivations during sleep. Memory consolidation during sleep has been shown to be preferential towards initially weaker memory traces and less supportive to memories that have already established robust representations (Drosopoulos et al., 2007). Since recall and recognition accuracy was lower for items acquired through incidental learning conditions, this is a compelling alternative explanation for why there was less forgetting of items learned through story reading.

In addition, the additional retrieval practice for the test immediately after training may have aided learning in the incidental, story reading condition. Further supporting this possibility, items learned through the incidental condition showed a slight improvement (8% increase) in cued recall accuracy between the test on day one and the test on day two. The second test task in the immediate test session (multiple-choice meaning-to-word matching) may therefore have boosted learning, manifesting as improved cued recall at the delayed test. The potential involvement of a testing effect (Roediger & Karpicke, 2006a) in long-term retention of new meanings for familiar words is examined in detail in Experiments 2 and 3.

Experiment 2: the testing effect in incidental and intentional learning

The aim of Experiment 2 was to investigate whether 24-h retention of novel word meanings that were learned through story reading can be boosted by introducing a test of participants’ knowledge immediately after training. As with Experiment 1, performance was compared to a more conventional explicit learning baseline (the same intentional training procedure as in Experiment 1). Participants were tested immediately after training on half of the items they saw in each learning condition, and then given a surprise test 24 h later in which they were tested on the other half of the items for the first time, as well as being retested on items that had been tested the previous day. As for Experiment 1, both test sessions consisted of cued recall of the new meanings, followed by a multiple-choice meaning-to-word matching test.

We predicted that there would be better long-term retention for items that were tested immediately after training, compared with those that were not tested, for both the incidental and intentional conditions. There was no specific prediction as to whether the magnitude of the testing effect would differ for the different learning conditions. Additionally, we expected to replicate the findings from Experiment 1 that retention would be better overall for novel meanings learned through intentional conditions, but that there would be less forgetting under incidental conditions.

Experiment 2 was preregistered through the Open Science Framework; the preregistration can be retrieved from https://osf.io/e5zmk (Hulme & Rodd, 2016, November 4). Any deviations from the preregistration are noted in the Method and Results sections for this experiment. The materials, data, and analysis scripts for Experiment 2 can be found on the Open Science Framework (OSF; https://osf.io/upmnr).

Method

Participants

We aimed to recruit 96 participants for Experiment 2 in which participants were trained on eight items (two items per cell: four per learning condition, and four per test type) in one of 16 experiment versions (six participants per version). The sample size was established in consideration of the study by Hulme, Barsky & Rodd (2019) and Experiment 1. As this experiment has the additional independent variable of test type, a larger sample size was used in Experiment 2 than Experiment 1 to achieve comparable power with a smaller number of items per cell.

Ninety-nine participants were included in the experiment (age: M = 32.31 years, SD = 8.14; 56 female); we accidentally over-recruited by three participants when pseudorandomly assigning participants to the experiment versions and kept these participants. Participants were recruited in the same way as for Experiment 1. They gave their informed consent before taking part (by means of ticking boxes in the online consent form) and were paid for their participation at the end of each session (£6 for session one and £2 for session two). The UCL Experimental Psychology Ethics Committee granted ethical approval for the research (Ref: EP/2017/009).

An additional 36 participants took part in the first session but did not complete session two by the deadline (within 6 h of receiving the invitation for the delayed test) and were excluded. (This additional data exclusion was used for Experiments 2 and 3 to ensure sufficient power to examine the testing effect within-participants, for which a complete set of immediate and delayed test data was critical.) A further twenty-one participants were excluded due to getting more than one of the multiple-choice comprehension questions wrong when reading the stories (see Experiment 1 Procedure), and two further participants were excluded for attempting the experiment more than once. Finally, five participants were excluded for being outliers in their mean reading speed (faster than 543.4 words per min, 2 SD above the mean). Excluded participants were replaced during recruitment.

Materials

The stimuli for the present experiment were identical to those used in Experiment 1.

Design

The experiment used a within-participants and within-items design, with two independent variables: learning condition (two levels: incidental and intentional) and test type (three levels: immediate test (tested in the first session), delayed test (tested for the first time in the second session), and delayed retest (tested for the second time in the second session)). The dependent variables were accuracy in cued recall of meanings and multiple-choice meaning-to-word matching.

There were sixteen versions of the experiment to ensure the items were seen an even number of times in each condition, with the order of the learning conditions counterbalanced across participants. Participants were pseudorandomly assigned to one of the sixteen versions of the experiment. As in Experiment 1, each participant was trained on half the total number of stimuli (eight items; four in each learning condition), with the items in each condition and the order of the learning conditions counterbalanced across participants. Further counterbalancing accounted for which stimulus items were or were not tested immediately following training across participants.

Procedure

The procedure for the incidental and intentional learning conditions was identical to that of Experiment 1, and immediately following training participants completed the same Mill Hill vocabulary test (Mill Hill Vocabulary Test, Set A: Multiple Choice: Raven, Raven & Court, 1998) as a filler task. Participants were then immediately tested on half of the items that they had been trained on through the incidental and intentional learning conditions (four items, two trained through each training method). The tests were the same as for Experiment 1: cued recall of meanings followed by multiple-choice meaning-to-word matching. The items were tested in a random order in each of the two tests, with no feedback given to participants. In the multiple-choice test only the four words that a participant was being immediately tested on appeared as the four alternative responses to choose from for each test item; the order of these was also randomised for each test item.

Exactly 24 h after the first session of the experiment had been made available, participants were asked to take part in the second session: the delayed test. Participants were not aware beforehand that they would be asked to complete this test to discourage them from rehearsing and intentionally retaining information about the novel word meanings. As such, unfortunately 36 participants did not return to complete session two; 99 participants were therefore included in the analysis. Participants began the delayed test an average of 24 h and 25 mins (SD = 57 mins, range = 22 h 45 mins–27 h 21 mins) after the training session. The tests were the same as those that had been used for the immediate test in the same order. This time participants were tested on all of the stimuli that they had been trained on (eight items). The order of presentation of the items in each of the two tests was again randomised for each participant, and for the multiple-choice test the order of the eight stimulus words to choose from was again randomised for each test item.

Results

Analysis procedure

Responses for the cued recall test and multiple-choice test were coded for accuracy (“1” for correct and “0” for incorrect) in the same way as for Experiment 1.

Upon completion of the experiment, we realised that test type was confounded with differing test difficulty between the immediate test and the two delayed test types for the multiple-choice meaning-to-word matching measure. This was because in the immediate test participants could choose from four alternative words to pair with the appropriate meaning on each trial (as they were only tested on half the total items they were trained on: four items, two trained through each training condition). In contrast, in the delayed test participants had to choose from eight alternatives (as they were tested on all eight of the items they had been trained on). The results from the immediate multiple-choice test are therefore not comparable to the results from the two delayed test types, so the analysis for this measure was only carried out on the subset of results for the two delayed test types. This is a deviation from the analysis plan outlined in the preregistration of this experiment. The analysis of the cued recall measure was carried out according to the preregistration.

The data were analysed, as in the previous experiment, using logistic mixed effects models with the lme4 package (version 1.1–12; Bates et al., 2015) and R statistical software (version 3.3.2, R Core Team, 2017), with two separate models for the analysis of data from the two measures. The model used to analyse the cued recall data contained three factors: test type (three levels: immediate, delayed first test, delayed second test), learning condition (two levels: incidental, intentional), and order of the learning conditions in the experiment (two levels: first, second). The contrasts for the fixed effect of test type were defined using Helmert coding, with one contrast comparing the immediate test to the two delayed tests combined (immediate: 0.67, delayed first test: –0.33, delayed second test: –0.33), and a second comparing the two delayed test types to each other (immediate: 0, delayed first test: –0.5, delayed second test: 0.5). Deviation coding was used to specify the contrasts for the fixed effects of learning condition (incidental: –0.5, intentional: 0.5) and learning order (first: –0.5, second: 0.5).

The model used to analyse the multiple-choice data also had three factors: test type (2 levels: delayed first test, delayed second test), learning condition (2 levels: incidental, intentional), and learning order (2 levels: first, second). The contrasts were specified using deviation coding for the fixed effects of test type (delayed first test: −0.5, delayed second test: 0.5), learning condition (incidental: −0.5, intentional: 0.5), and learning order (first: −0.5, second: 0.5). The procedure for determining the appropriate random effects structure and significance of the fixed effects/interactions was the same as described for Experiment 1. The model used to analyse the multiple-choice data used the maximal random effects structure; the one for the cued recall measure was simplified by removing the correlations between the by-participant and by-item random slopes and random intercepts (as recommended by Barr et al., 2013).

Following on from the main analysis for the cued recall measure, firstly three pairwise comparisons (with Bonferroni adjustment for multiple comparisons, α = .017) were carried out to compare the different levels of test type to each other. This was done by taking a subset of the data for each pair of levels of test type and creating a model for each containing the same fixed and random effects as the model used for the main analysis, although the contrast for test type was coded using deviation coding (immediate: 0.5, delayed first test: −0.5; immediate: 0.5, delayed second test: −0.5; delayed first test: −0.5, delayed second test: 0.5). Secondly, further follow-up pairwise comparisons (with Bonferroni correction for multiple comparisons, α = .017) were made for the three 2 × 2 interactions between the pairs of test types and the two learning conditions to determine whether the difference between any two test types differed between the two learning conditions. Finally, six simple effects pairwise comparisons (with Bonferroni adjustment for multiple comparisons, α = .008) were run to test for any significant differences between the different test types within the two learning conditions. This was done by taking further subsets of the data for the pairs of levels of test type separately for the incidental and intentional learning conditions and creating models with only fixed effects for test type, learning order, and the interaction (with random effects for test type for participants and items).

The only follow-up analyses carried out for the multiple-choice test were two simple effects pairwise comparisons (with Bonferroni correction for multiple comparisons, α = .025). This was done in the same way as the simple effects analyses for the cued recall measure. All data and analysis scripts for this experiment are available via the Open Science Framework (OSF; https://osf.io/upmnr).

Cued recall of meanings

Data for the cued recall test (Fig. 3) showed that accuracy was significantly higher for items trained through the intentional than the incidental learning condition (χ2(1) = 34.83, p < .001). There was also a significant main effect of test type (χ2(2) = 25.78, p < .001); the main effect of learning order was non-significant (χ2(1) = 2.50, p = .114). There was a significant interaction between learning condition and test type (χ2(2) = 13.86, p < .001), but the interaction between learning condition and learning order was not significant (χ2(1) = 0.09, p = .760). There was an unexpected significant interaction between test type and learning order (χ2(2) = 9.24, p = .010); the three-way interaction was not significant (χ2(2) = 3.23, p = .199) (Results for all the analyses for the cued recall test are also presented in Table 1 for clarity of the different levels of follow-up analysis for this measure).

Figure 3 Experiment 2. Mean percentage of correct responses in the cued recall test for each learning condition and for the three different test types in the experiment.

Error bars show standard error of the means adjusted for the within-participants design (Cousineau, 2005).

Table 1 Results for the linear mixed effects model analyses for the cued recall test in Experiment 2 showing the main analyses and the different levels of follow-up analyses.

Fixed effect or interaction	χ 2	df	p	
1. Main analysis (α = .05)	
Learning condition (Incidental vs. Intentional)	34.83	1	<.001	
Test type (Immediate vs. Delayed 1st vs. Delayed 2nd)	25.78	2	<.001	
Learning order (First task vs. Second task)	2.50	1	.114	
Learning condition × Test type	13.86	2	<.001	
Learning condition × Learning order	0.09	1	.760	
Test type × Learning order	9.24	2	.010	
Learning condition × Test type × Learning order	3.23	2	.199	
2. Follow-up pairwise comparisons of the different levels of Test type (α = .017)	
Immediate vs. Delayed 1st tests	19.99	1	<.001	
Delayed 1st vs. Delayed 2nd tests	18.83	1	<.001	
Immediate vs. Delayed 2nd tests	0.89	1	.345	
3. Follow-up pairwise comparisons of 2 × 2 interactions between pairs of levels of Test type and the two Learning conditions (α = .017)	
Learning condition × Immediate vs. Delayed 2nd tests	16.24	1	<.001	
Learning condition × Immediate vs. Delayed 1st tests	2.29	1	.130	
Learning condition × Delayed 1st vs. Delayed 2nd tests	2.97	1	.085	
4. Simple effects analyses of differences between levels of Test type within the two Learning conditions (α = .008)	
Immediate vs. Delayed 1st tests for Incidental learning	5.88	1	.015	
Delayed 1st vs. Delayed 2nd tests for Incidental learning	15.27	1	<.001	
Immediate vs. Delayed 2nd tests for Incidental learning	14.59	1	<.001	
Immediate vs. Delayed 1st tests for Intentional learning	18.27	1	<.001	
Delayed 1st vs. Delayed 2nd tests for Intentional learning	8.39	1	.004	
Immediate vs. Delayed 2nd tests for Intentional learning	4.25	1	.039	
Exploratory analyses: Pairwise comparisons of 2 × 2 interactions between pairs of levels of Test type and the two Learning orders (α = .017)	
Learning order × Delayed 1st vs. Delayed 2nd tests	7.12	1	.008	
Learning order × Immediate vs. Delayed 2nd tests	0.12	1	.729	
Learning order × Immediate vs. Delayed 1st tests	5.15	1	.023	
Exploratory analyses: Simple effects analyses of differences between levels of Test type within the two Learning orders (α = .025)	
First task vs. Second task for Immediate test	2.25	1	.134	
First task vs. Second task for Delayed 1st test	2.04	1	.154	
First task vs. Second task for Delayed 2nd test	4.91	1	.027	
Note:

Test results reported are for likelihood ratio tests comparing the full model to models with each fixed factor/interaction of interest removed in turn.

To further investigate the significant main effect of test type, three pairwise comparisons between the different levels of test type were carried out. The results revealed that there was a significant effect of overnight forgetting (difference between the delayed 1st and immediate tests) (χ2(1) = 19.99, p < .001), with better recall of new meanings when tested immediately than when tested for the first time after a delay. There was also a significant testing effect (difference between the delayed 2nd and delayed 1st tests) (χ2(1) = 18.83, p < .001), with higher recall accuracy for items that were being tested for the second time than for those being tested for the first time after the delay. However, there was no significant difference in cued recall accuracy between the immediate and delayed 2nd tests (χ2(1) = 0.89, p = .345; α = .017), suggesting these items were protected against forgetting.

To further investigate the significant interaction between learning condition and test type, the second set of follow-up analyses were pairwise comparisons for the three 2 × 2 interactions between the pairs of test types and the two learning conditions. Results showed that there was a significant interaction between learning condition and the difference between the immediate and delayed 2nd tests (χ2(1) = 16.24, p < .001). Items learned incidentally from stories showed some improvement between the immediate test and the retest on day two, while items learned through the intentional learning condition showed a small amount of forgetting. There was no significant interaction between learning condition and the difference between either the immediate and delayed 1st tests (χ2(1) = 2.29, p = .130) or between the delayed 1st and delayed 2nd tests (χ2(1) = 2.97, p = .085; α = .017).

In the final set of follow-up analyses, six simple effects pairwise comparisons were run to test for any significant differences between the different test types within the two learning conditions. The results revealed that for the incidental learning condition there was no significant difference in recall accuracy for items tested for the first time after the delay than for items tested immediately after training at the corrected level (χ2(1) = 5.88, p = .015), that is no significant forgetting. There was significantly better cued recall accuracy for items tested for the second time after the delay than those tested for the first time (χ2(1) = 15.27, p < 0.001): a significant testing effect. There was also significantly better recall of new meanings tested for the second time after the delay than the immediate test (χ2(1) = 14.59, p < 0.001). For the intentional learning condition, there was significantly lower recall accuracy for items tested for the first time after the delay than those tested immediately (χ2(1) = 18.27, p < .001), showing significant forgetting. There was better recall accuracy for items tested for the second time after the delay than those tested for the first time (χ2(1) = 8.39, p = .004): a significant testing effect. There was no significant difference (at the corrected level) between items tested for the second time following the delay and when tested immediately after training (χ2(1) = 4.25, p = .039; α = .008).

Additionally, although not specified in the preregistration, exploratory follow-up analyses were carried out to examine the nature of the unexpected interaction between test type and learning order. These were three pairwise comparisons of the 2 × 2 interactions between the pairs of test types and the two learning orders (first or second position in the experiment). Results revealed a significant interaction between position and the difference between the delayed 1st and delayed 2nd tests (χ2(1) = 7.12, p = .008). Items appeared to be recalled better at the delayed 1st test when they had been presented in the first condition in the training session, whereas items were recalled better at the delayed 2nd test when they had been trained in the second condition. There was no significant interaction between the immediate and delayed 2nd tests (χ2(1) = 0.12, p = .729), nor between the immediate and delayed 1st tests (χ2(1) = 5.15, p = .023) at the Bonferroni-corrected level (α = .017). However, further follow-up analyses of the simple effects pairwise comparisons of learning order within the delayed 1st and delayed 2nd test types were both non-significant at the Bonferroni-corrected level (both p > .025).

Multiple-choice meaning-to-word matching

The analysis of the data for the multiple-choice meaning-to-word matching test (Fig. 4) was only carried out on the subset of results for the two delayed test types. Overall accuracy was very high, and it was significantly higher for items trained through the intentional condition than the incidental condition across all test types (χ2(1) = 8.44, p = .004). The main effect of test type was also significant (χ2(1) = 6.71, p = .010), with slightly greater accuracy for items that had been tested previously than for those that had not been; there was no significant main effect of learning order (χ2(1) = 0.32, p = .569). The interaction between learning condition and test type was not significant (χ2(1) = 0.61, p = .435), nor was the interaction between learning condition and learning order (χ2(1) = 1.23, p = .268), nor the interaction between test type and learning order (χ2(1) = 1.32, p = .251). The three-way interaction was also not significant (χ2(1) = 0.06, p = .810).

Figure 4 Experiment 2. Mean percentage of correct responses in the multiple-choice test for each learning condition and for the three different test types in the experiment.

Note that the results from the immediate test (with lighter shading) are not comparable to those from the two delayed test types due to an underlying difference in test difficulty. Error bars show standard error of the means adjusted for the within-participants design (Cousineau, 2005).

Following on from the main analysis, two simple effects pairwise comparisons tested for any significant differences between the different test types within the two learning conditions. In the incidental learning condition, the difference between items tested for the second time following the delay and those tested for the first time was non-significant at the corrected level (χ2(1) = 4.62, p = .032). For the intentional learning condition, there was also no significant difference in accuracy between items tested for the first or second time after the delay (χ2(1) = 3.53, p = .060; α = .025).

Discussion

The aim of Experiment 2 was to examine whether testing memory immediately after training enhances long-term retention of new word meanings acquired through incidental and intentional learning conditions, and to see whether any testing effect differs depending on the learning conditions. As in Experiment 1, new word meanings were learned better overall through intentional learning conditions than through incidental learning conditions. Accuracy in both the cued recall and multiple-choice tests was significantly higher for items trained through the intentional learning condition.

Cued recall accuracy was also higher overall immediately after training than when items were tested for the first time after 24 h. This demonstrates some overnight forgetting of the new meanings for the words in the absence of an intervening test. Furthermore, there was numerically but not significantly (at the corrected level) more forgetting of items trained through the intentional learning condition than those learned through a story, both with and without prior retrieval practice. This is in line with the findings of Experiment 1, where accuracy on the multiple-choice test was lower after 24 h for items learned through the intentional condition, but not for items learned through the stories.

Critically, both the cued recall and multiple-choice tests revealed an overall testing effect: new word meanings were recalled and recognised significantly better after 24 h when they had been tested immediately after training than when they were being tested for the first time. As predicted, in the cued recall measure this main effect of testing was also significant in the simple effects analyses that looked at incidental and intentional learning separately. This is in line with studies that have found a benefit of prior retrieval on learning information from different contexts, such as list of foreign language vocabulary words and their translations (Van den Broek et al., 2013) and information from prose passages (Roediger & Karpicke, 2006b).

The lack of difference in cued recall accuracy between performance on the immediate test and the delayed test for items being tested for the second time suggests that the retrieval practice protected these items against forgetting. The testing effect seen in the present study therefore at least partly explains why participants in Experiment 1 and those in the study by Hulme, Barsky & Rodd (2019) showed such good retention after one day and one week respectively. In the present experiment both the cued recall and multiple-choice tests were administered to all participants at both time points. It is therefore unclear whether either of these tests on its own would produce a similar testing effect, or if the combination of the two was important for boosting long-term retention. It also remains to be seen whether one of these test types is better than the other for enhancing retention of new word meanings learned incidentally through reading.

In sum, Experiment 2 demonstrated that testing memory of new meanings for familiar words benefits their future retention. This was the case for recalling word meanings learned either incidentally through story reading or through an intentional learning condition. As in Experiment 1, participants learned vocabulary more efficiently through the intentional learning condition, but performance for both learning conditions was good. There was a trend in the data towards less forgetting of items trained incidentally through the stories, and a trend suggesting a larger testing effect for incidentally-trained items. However, these interactions were non-significant; further research is warranted to investigate whether incidentally-trained items in particular could benefit from the additional learning opportunity afforded by the immediate test. Either the immediate cued recall or multiple-choice test, or indeed a combination of the two, may have produced the observed testing effect; Experiment 3 investigates which of these test methods could be more beneficial for retention. The results of this experiment will guide future development of real-world intervention studies aimed to boost vocabulary learning from story reading.

Experiment 3: immediate test method

The testing effect has been observed in studies using various methods of immediate test, most usually with cued recall (e.g., Karpicke & Smith, 2012), but also with other methods such as multiple-choice (e.g., Roediger & Marsh, 2005). There are several possibilities as to why certain methods of immediate testing may be more beneficial for future retention. The retrieval effort hypothesis states that testing is more helpful for long-term retention when it is more effortful (Pyc & Rawson, 2009). For example, in a study in which young adults learned the meanings of novel L1 vocabulary words, Karpicke & Roediger (2007) showed that increasing retrieval difficulty by increasing the delay between initial study and initial testing led to better long-term retention than when initial retrieval effort was lower. Tests of productive vocabulary knowledge, such as cued recall of word meanings, are more difficult than recognition tests in which word meanings are supplied (Pellicer-Sánchez, 2016); therefore an immediate cued recall test may be more advantageous for future retention than a multiple-choice test. Indeed, findings from many studies (for a review see Rowland, 2014) suggest effortful processing to be an important attribute of the testing effect.

Conversely, immediate testing may be particularly beneficial when it assists with restructuring learned information into a format that is more helpful for long-term retention. Multiple-choice recognition tests may aid retention due to response choices cueing the retrieval of marginal knowledge that may otherwise not be easily accessible (Marsh et al., 2007). They may also provide an opportunity for additional learning of some items through the process of elimination of foils (Marsh et al., 2007) even in the absence of feedback on response choice. However, foil answers in multiple-choice tests may also lead to learning of incorrect information (Butler et al., 2006; Marsh et al., 2007; Roediger & Marsh, 2005).

Some studies have directly compared the effects of immediate cued recall and multiple-choice tests on long-term retention. Duchastel (1981) investigated secondary school students’ retention of a prose passage following immediate testing with either a short-answer test (akin to cued recall), a multiple-choice test, or a free recall test. Long-term retention (measured by a delayed cued recall test) was better for those who had the immediate short-answer test, but no testing effect was observed for the other two groups. However, Duchastel (1981) found no testing effect for any group on the delayed free recall test, and the delayed cued recall measure was very similar to the immediate test for the short-answer test group. More recently, Nakata (2016) compared retrieval methods including cued recall and multiple-choice recognition in a study of paired-associate learning of novel L2 words. Recall was found to be most beneficial for acquiring novel words’ orthography (spelling), whereas recognition was more beneficial otherwise (Nakata, 2016).

One concern is that information learned with the help of retrieval practice could be relatively inflexible and constrained, and may therefore not transfer to different delayed tests. Tran, Rohrer & Pashler, 2014 and others have found that retrieval practice may not benefit later tests that require making deductive inferences about the learned information. Furthermore, Hogan & Kintsch (1971) found that immediate test methods that provide further exposure (i.e., recognition tests) were more beneficial than free recall for recognition two days later, whereas both free recall and recognition boosted performance on delayed free recall.

However, the degree to which different methods of immediate test aid future retention can also differ depending on factors such as the provision of feedback. Kang et al. (2007) found that participants who had an immediate multiple-choice test performed better on delayed multiple-choice and short-answer tests than participants who had an immediate short-answer test (Experiment 1; Kang et al., 2007). However, in a second experiment where feedback was provided on initial test performance (Experiment 2; Kang et al., 2007), the group with the immediate short-answer test performed better on the delayed tests than those whose immediate test had been multiple-choice, supporting the retrieval effort hypothesis (Pyc & Rawson, 2009). Other studies have also found that the testing effect can transfer across different test methods (Butler, 2010; McDaniel et al., 2007; Rohrer, Taylor & Sholar, 2010), with cued recall usually found to be more beneficial for long-term retention than recognition tests.

The aim of Experiment 3 was to investigate the impact of immediate test method (cued recall vs. multiple-choice meaning-to-word matching) on retention of new word meanings learned through stories. A secondary aim was to rule out the possibility that the testing effect in Experiment 2 was simply a practice effect due to having previously completed the same test, and that retrieval practice can generalise to a different delayed test. We predicted that the results would replicate the key finding from Experiment 2 of better long-term retention for items tested immediately after training (regardless of testing method) than items not tested previously. Additionally, we predicted that cued recall would be more beneficial for long-term retention of new word meanings than multiple-choice meaning-to-word matching as, according to the retrieval effort hypothesis (Pyc & Rawson, 2009), production tests that require more effortful retrieval than recognition tests (Roediger & Butler, 2011; Rowland, 2014) are more helpful for retention. However, it was also possible that the multiple-choice meaning-to-word matching test could result in better subsequent retention as this test provides participants with additional cues that provide an additional learning opportunity (Marsh et al., 2007). Finally, it was possible that the testing effect would not transfer across test tasks (Hogan & Kintsch, 1971; Tran, Rohrer & Pashler, 2014), and so the benefit of each method of immediate test would only be seen for the delayed test of the same type, in which case it could be characterised as more of a practice effect.

Experiment 3 was preregistered through the Open Science Framework; the preregistration is available at https://osf.io/c59tz (Hulme & Rodd, 2017, June 23). Any deviations from the preregistration are noted in the Method and Results sections for this experiment. The materials, data, and analysis scripts for Experiment 3 can be found on the Open Science Framework (OSF; https://osf.io/eh2c6).

Method

Participants

We aimed to recruit 96 participants for Experiment 3 in which participants were trained on eight items (four per testing condition) in one of two groups (who had different immediate tests), with eight experiment versions (12 participants per version). The sample size was established in consideration of the previous experiments. While there were more items per cell per participant in Experiment 3 than Experiment 2, the additional between-participants factor of test method meant that a similarly large sample was required to achieve comparable power in this experiment.

Ninety-eight participants were included in the experiment (age: M = 33.7 years, SD = 8.0; 64 female); we over-recruited by two participants when pseudorandomly assigning participants to the experiment versions and kept these participants. Participants were recruited in the same way as for Experiments 1 and 2. They gave their informed consent before taking part (by means of ticking boxes in the online consent form) and were paid for their participation at the end of each session (£4 for session one and £2 for session two). The UCL Experimental Psychology Ethics Committee granted ethical approval for the research (Ref: EP/2017/009).

An additional 18 participants took part in the first session but did not complete session two by the deadline (within 6 h of receiving the invitation for the delayed test) and were excluded. A further thirty-five participants were excluded due to getting more than one of the multiple-choice comprehension questions wrong in either of the stories they read (see Experiment 1 Procedure). Seven further participants were excluded due to a technical issue during data collection, and two participants were excluded for being outliers in their mean reading speed (faster than 806.2 words per min, 2 SD above the mean). Excluded participants were replaced during recruitment.

Materials

The stimuli for Experiment 3 were identical to those used in the previous experiments. One additional different paraphrased version of each of the definition sentences was created so that a differently worded definition would be presented in the immediate and delayed multiple-choice meaning-to-word matching tests to counteract any direct practice effects (see Table S3 for the additional paraphrased definitions: https://osf.io/tnb94).

Design

The experiment used a mixed design with two independent variables: immediate test method (two levels: cued recall and meaning-to-word matching) was manipulated between participants, and whether items were or were not previously tested (two levels: not previously tested vs. previously tested) was manipulated within subjects. The dependent variables were accuracy in cued recall of the new meanings and multiple-choice meaning-to-word matching, measured at the delayed test time point.

There were eight versions of the experiment to ensure the stimuli were seen an even number of times in each condition across participants. Participants were pseudorandomly assigned to one of the eight versions of the experiment. As in Experiments 1 and 2, each participant was trained on half the total number of stimuli (eight items per participant; two separate stories). For the key factor of immediate test method, half of the participants (N = 49) were given a cued recall test of half of their items (four items) immediately after training, and the other half of the participants (N = 48) were given a multiple-choice meaning-to-word matching test of half of their items immediately after training. The items that were or were not tested immediately following training were also counterbalanced across participants.

Procedure

The first session of the experiment began with the incidental training procedure. Participants first read one of the short stories; the procedure for this was identical to that of the previous experiments. They were then asked to rate how enjoyable and clear they found the story, and answer some questions about their subjective reading style, which took around 2 mins—this served as a brief interval between the two stories. Participants then read a second story. Immediately following training participants completed the same Mill Hill vocabulary test as used in the previous experiments (Mill Hill Vocabulary Test, Set A: Multiple Choice: Raven, Raven & Court, 1998) as a filler task. Participants were then given an immediate test of half the items they had been trained on (four items, two from each story), which was either a cued recall test or a multiple-choice meaning-to-word matching test. The items were tested in a randomised order in both of the test tasks, with no feedback given to participants. In the multiple-choice meaning-to-word matching test, only the four words that a participant was being immediately tested on appeared as the four alternative responses to choose from for each test item; the order of hese was also randomised for each test item.

Exactly 24 h after the first session of the experiment had been made available, the participants were asked to take part in the second session of the experiment: the surprise delayed test. Again participants were not aware of this delayed test beforehand; unfortunately 18 participants did not return to complete session two and were replaced during data collection. Participants completed the delayed test an average of 24 h and 31 mins (SD = 57 mins; range = 22 h 40 mins–27 h 25 mins) after the training session. The tests used for the delayed test session were the same as for the immediate test, but this time participants completed both tests: cued recall of meanings followed by multiple-choice meaning-to-word matching. At the delayed test participants were tested on all of the stimuli that they had been trained on (eight items). The order of presentation of the items in each of the two tests was again randomised for each participant. For the multiple-choice test, different paraphrased versions of the definition sentences were used to those that had appeared in the immediate test, and the order of the eight words to choose from was randomised for each test item.

Results

Analysis procedure

Responses on the cued recall and multiple-choice tests were coded for accuracy in the same way as for the previous experiments. The data were again analysed using logistic mixed effects models with the lme4 package (version 1.1–13; Bates et al., 2015) and R statistical software (version 3.3.3; R Core Team, 2017). Two models were created to analyse the results of the two delayed tests of cued recall and multiple-choice meaning-to-word matching separately. Contrasts for the fixed effects were defined using deviation coding for whether items were or were not immediately tested (not previously tested: –0.5, previously tested: 0.5), and the immediate test method (cued recall: –0.5, multiple-choice: 0.5), with the interaction coded by multiplying the contracts for these two factors. The model for the cued recall data used the maximal random effects structure, with by-participant and by-item random intercepts, a by-participants random slope for whether items were or were not previously tested, and by-items random slopes for whether items were or were not previously tested, method of immediate test, and the interaction. The model for the multiple-choice measure was simplified by removing the correlations between the random slopes and random intercepts (as recommended by Barr et al., 2013).

Following on from the main analysis, simple effects analyses were carried out to determine significance of whether items had or had not been immediately tested within each of the two immediate test methods separately. All data and analysis scripts for this experiment are available via the Open Science Framework (https://osf.io/eh2c6).

Cued recall of meanings

The accuracy data for cued recall (measured in the delayed test session) are shown in Fig. 5. Cued recall performance was low overall when items had not been tested immediately after training and was similar for the immediate cued recall group (26.5%) and immediate multiple-choice meaning-to-word matching group (25.5%): there was no significant main effect of immediate test method (χ2(1) = 0.47, p = .491). This is reassuring as it shows that the two groups of participants (who had different immediate test methods) performed similarly overall. There was a significant main effect of whether items were or were not immediately tested (χ2(1) = 23.73, p < .001): performance was much higher when items had been tested immediately after training. Although accuracy appeared higher for the immediate multiple-choice group (58.3%) than for the immediate cued recall group (49.5%) when items had been previously tested, the interaction was non-significant (χ2(1) = 3.18, p = .074). The planned simple effects follow-up analysis showed that there was a significant effect of whether items were or were not immediately tested within the immediate cued recall group (χ2(1) = 8.25, p = .004), and also within the multiple-choice meaning-to-word matching group (χ2(1) = 25.10, p < .001; α = .025).

Figure 5 Experiment 3. Mean percentage of correct responses in the cued recall test measured at the delayed test for participants whose immediate test was cued recall, and for those whose immediate test was multiple-choice when items were or were not previously tested.

Error bars show standard error of the means adjusted for the within-participants factor of whether items were or were not previously tested (Cousineau, 2005).

Multiple-choice meaning-to-word matching

The accuracy data for the multiple-choice meaning-to-word matching test (measured in the delayed test session) are shown in Fig. 6. Performance on this test was much higher overall than on the cued recall test. Accuracy was significantly lower when items had not been tested immediately after training (χ2(1) = 14.54, p < .001), and there was no significant main effect of immediate test method (χ2(1) = 2.38, p = .123). Concerning a possible interaction between whether items were or were not immediately tested and the immediate test method, the simple means for accuracy were similar for the immediate cued recall group (60.5%) and the immediate multiple-choice meaning-to-word matching group (63.5%) when there was no previous test. This is again reassuring as it shows that the two groups of participants (who had different immediate test methods) performed similarly overall. Although the simple means for accuracy on items that had been tested previously appeared to be higher for the immediate multiple-choice group (78.6%) than for the immediate cued recall group (67.5%), the interaction was non-significant (χ2(1) = 2.48, p = .116). The planned simple effects follow-up analysis showed that the effect of whether items were or were not immediately tested was non-significant within the immediate cued recall group (χ2(1) = 3.31, p = .069). The effect was, however, significant within the multiple-choice meaning-to-word matching group (χ2(1) = 10.76, p = .001; α = .025).

Figure 6 Experiment 3. Mean percentage of correct responses in the multiple-choice test measured at the delayed test for participants whose immediate test was cued recall, and for those whose immediate test was multiple-choice, when items were or were not previously tested.

Error bars show standard error of the means adjusted for the within-participants factor of whether items were or were not previously tested (Cousineau, 2005).

Discussion

Experiment 3 replicated the key finding from Experiment 2: testing memory immediately after training significantly boosted retention of new word meanings learned incidentally through story reading as measured at the delayed tests 24 h later. Importantly, Experiment 3 showed that testing memory immediately after training using either cued recall or multiple-choice alone was sufficient to result in a significant testing effect. This is consistent with studies that have found a testing effect arising from an immediate cued recall test (Karpicke & Smith, 2012) or an immediate test using multiple-choice questions (Roediger & Marsh, 2005).

The results provide no strong evidence for a particular benefit for either of these two testing methods: the retention benefits following the immediate multiple-choice test were non-significantly larger than the immediate cued recall test. The simple effects showed that the immediate multiple-choice test significantly boosted performance on both of the delayed tests, while the immediate cued recall test only significantly enhanced performance on the delayed cued recall test but not on the delayed multiple-choice test (the effect was marginally significant at the uncorrected level).

Finally, the immediate multiple-choice test enhanced delayed cued recall of the new word meanings. This cross-task transfer effect suggests that the observed benefit is not simply due to practising the same test previously. This demonstrates that knowledge retained from prior testing can be flexibly applied to new contexts of retrieval, in line with the findings of Rohrer, Taylor & Sholar (2010) and others.

General discussion

The present experiments investigated participants’ ability to learn new word meanings from naturalistic story contexts. Specifically we (i) compared performance in this relatively incidental learning condition to a more conventional explicit learning paradigm and (ii) examined the role of testing memory after training in enhancing future retention of new word meanings.

Incidental versus intentional learning

In Experiments 1 and 2 participants learned new meanings for familiar words better under intentional learning conditions than incidentally through reading stories. These findings are in line with those of studies that have compared incidental and intentional learning in studies of L2 vocabulary learning (Hulstijn, 1992; Peters et al., 2009) and L1 vocabulary learning with adolescents (Konopak et al., 1987).

The intentional and incidental learning paradigms differed in several important ways, making it difficult to draw firm conclusions for the specific underlying cause of this difference. It may be driven by additional attentional focus on word meanings in the intentional learning condition, while in the incidental learning condition participants’ attention was directed towards other aspects of the rich narrative context of the stories. The two learning paradigms also differed in the spacing of the words, which were systematically spaced throughout the intentional learning task. In the stories, on the other hand, the new word meanings appeared at naturally-occurring intervals such that some exposures occurred relatively close together. Spacing stimuli apart has been widely shown to aid learning (for review see: Dempster, 1996).

Additionally, there is the possibility of an internal testing effect within the intentional learning task. The multiple-choice meaning-to-word matching portion of the intentional learning task was similar to the multiple-choice meaning-to-word matching task used in the testing phase, which produced a testing effect on its own in Experiment 3. The multiple-choice training task also included simple feedback on performance (“correct” or “incorrect”), and feedback can enhance the benefit of tests for future retention. The trend in the data of Experiment 2 towards a larger testing effect for the incidental learning condition could possibly be because the intentional learning condition already involved a limited testing effect.

As well as the overall differences in performance between learning conditions immediately after training, there were also differences in longer-term retention. After 24 h, participants in Experiment 1 (and non-significantly in Experiment 2) had forgotten some of the new word meanings learned under intentional conditions, but there was very little forgetting of items learned incidentally across both experiments. In Experiment 2 the trend for reduced forgetting of items learned under incidental conditions was present regardless of an immediate test. The significant interaction between day and learning condition in Experiment 1 provides evidence for a difference in the amount of forgetting: word meanings learned incidentally, although harder to learn initially, may be forgotten less quickly than those learned under intentional conditions. This is possibly due to the more semantically rich context of the stories providing participants with additional and more varied cues, which are advantageous for later retrieval of the new word meanings, or due to higher engagement as the stories are more enjoyable. An alternative possibility is that this difference is a function of the initial learning level as performance in the intentional learning condition was higher and so had further to fall. Another possibility is that memory consolidation during sleep may have played a role in preferentially strengthening the weaker memory traces of items acquired through incidental conditions (Drosopoulos et al., 2007). Nevertheless, while intentional learning conditions were better for more efficient acquisition, incidental learning may lead to less forgetting of word meanings over time. However, replication of the latter finding is warranted in future research in a design that matches on initial performance level between the different learning conditions. This may also help to determine whether the observed changes in performance over time reflect real differences in forgetting or differences in the two conditions in terms of different sensitivity to recall thresholds following the immediate test (see Kornell, Bjork & Garcia, 2011 for a discussion). Future replication of this finding is also important given the non-significant interaction between learning condition and immediate versus delayed first test in Experiment 2.

The testing effect

A large overall testing effect was found in both Experiments 2 and 3: retrieval practice following initial exposure boosted retention of new meanings for familiar words. Cued recall of the new word meanings was boosted by 28.8% for the incidental learning condition and 10.6% for the intentional learning condition in Experiment 2. This effect therefore likely explains the high levels of cued recall and multiple-choice accuracy in Experiment 1 after one day, and in Hulme, Barsky & Rodd (2019)’s study after seven days. This finding adds to the growing literature highlighting the role of testing in aiding vocabulary learning. A possible alternative explanation for this finding is that testing may bias participants towards selectively remembering tested items, so the effect may be driven by the cost to the untested items. Relatedly, research has shown that the retention of selected memories can be modulated after learning by giving simple verbal instructions on their future importance (Van Dongen et al., 2012). However, several studies in the literature using a between-subjects design (e.g., Experiment 2 of Roediger & Karpicke, 2006a) have demonstrated that the practice of testing enhances learning, rather than just biasing which items are remembered.

The testing effect in Experiment 2 was elicited with both a cued recall and multiple-choice test immediately after training. However, the findings of Experiment 3 illustrated that either a cued recall test or a multiple-choice test alone was sufficient to produce a significant testing effect following incidental learning through story reading. There was no clear evidence that either of the two testing methods was superior to the other for enhancing memory retention, although the retention benefits for the immediate multiple-choice test were non-significantly larger than for the cued recall test. Previous research has shown cued recall (short answer questions) to be more helpful for memory retention in some contexts (Duchastel, 1981; McDaniel et al., 2007; McDaniel, Roediger & McDermott, 2007). However, recent research has suggested that the relative benefit of the testing method may depend on the type of information being learned. Nakata (2016) found that recall was most helpful for acquiring novel words’ orthography (spelling), whereas recognition was more beneficial otherwise.

Despite the growing body of research on the benefits of retrieval practice for retention, the neurocognitive mechanisms underlying the testing effect remain somewhat unclear (Antony et al., 2017). Influential models of word learning have not yet provided an account of the testing effect. The Complementary Learning Systems (CLS) model of word learning, for example, describes how word forms are initially encoded into episodic memory in the hippocampus, and are integrated into semantic memory in the neocortex following a period of offline consolidation, such as during sleep (Davis & Gaskell, 2009). However, this model does not account for the effect of conscious retrieval on memory for new words and their meanings. Antony et al. (2017) recently suggested that a similar mechanism may underlie both offline consolidation and the testing effect, which may provide a fast track to consolidation. They argue that retrieval practice brings about the formation of flexible hippocampal-neocortical representations through the online reactivation of related knowledge (Antony et al., 2017). The testing effect is therefore important to consider in conjunction with offline consolidation processes to garner a full picture of how novel word meanings are remembered. Future research should explore whether the testing effect involves a similar mechanism as unconscious offline learning processes, thus providing a fast track to consolidation.

The findings of the present experiments have important methodological implications for studies of word learning. The enhancing effects of retrieval practice on memory are clearly shown here, and in other previous research. Studies considering the impact of factors such as the role of sleep for consolidation should (and do) consider this important aspect. For example, Henderson et al. (2015) compared adults’ and children’s explicit memory of new words using cued recall and recognition tests administered both immediately and 24 h later. They note that for both adults and children “explicit phonological memory was enhanced after off-line consolidation” p.413 (Henderson et al., 2015). However, this finding could also, at least in part, be attributable to a testing effect, as the 24-h tests repeated the 0-h tests that participants completed immediately after training. Nevertheless, other studies in this field have isolated effects of sleep from effects of testing using alternative designs, including train twice, test once (e.g., Weighall et al., 2017), and AM-PM designs to compare 12-h periods associated with wake and sleep (e.g., Henderson et al., 2012; James, Gaskell & Henderson, 2020). The testing effect is therefore an important consideration for studies of the cognitive mechanisms underlying vocabulary learning and retention that include repeated testing of trained words. Studies of sleep and vocabulary learning would benefit from using designs that avoid having multiple test sessions. For example, training different items at different times and testing all items in one final session (e.g., Experiment 1 of Tamminen & Gaskell, 2013), or using a between-groups design to compare between two different times of test (e.g., Experiment 1 of Tamminen, Davis & Rastle, 2015) avoids contaminating results of potential consolidation with those of a testing effect.

The testing effect has previously been shown to generalise to educational settings. For example, Roediger et al. (2011) found that repeated testing of real course content with multiple-choice (and some short-answer tests) successfully boosted middle school students’ grades on their social studies course. A similar study with a middle school science class (McDaniel et al., 2011) found that multiple-choice tests gave large gains (13–25%) in learning and retention, assessed by end-of-unit exams, especially when tests were taken closer to exam time. Similar benefits of retrieval practice have been seen for college students (McDaniel et al., 2007; McDaniel, Roediger & McDermott, 2007), and Larsen, Butler & Roediger, 2008 have advocated the use of test-enhanced learning in medical education. Our findings in Experiments 2 and 3 have important practical implications for vocabulary learning. Students learning vocabulary incidentally from reading storybooks or textbooks can benefit from being tested following initial encounters with new word meanings. Testing appears to be effective using either cued recall or multiple-choice methods, so incorporating it as part of a strategy for efficient vocabulary learning could be easy to implement. Tests are often considered solely as tools to assess learning, however they also provide an important opportunity for additional learning and reinforcement of knowledge.

Conclusions

The first two experiments confirmed that new word meanings are learned more efficiently under intentional learning conditions than incidentally through story reading. However, there was also some evidence of less forgetting of items learned through stories, suggesting that word meanings learned in a more semantically rich context could be retained better. The second two experiments showed that testing memory aids retention of new word meanings acquired under either incidental or intentional learning conditions. Both cued recall and recognition (multiple-choice) tests enhanced retention, but multiple-choice tests gave non-significantly better performance, even with no feedback. Furthermore the testing effect transferred across test tasks: immediate multiple-choice meaning-to-word matching improved accuracy on the delayed cued recall test, so the effect is not restricted to benefitting the same test.

The present study has demonstrated that testing memory following initial exposure is a powerful way to improve learning and long-term retention of vocabulary knowledge. Importantly, we found that retrieval practice benefitted vocabulary retention from different learning conditions and using different methods of immediate test. The vast majority of new vocabulary is learned through reading from mid-childhood onwards. Test-enhanced learning could therefore be particularly useful if implemented during vocabulary development to boost children’s vocabulary gains from story reading.

The authors would like to thank Helen Moss and Johan Heemskerk for authoring the stories used as stimulus materials, and Rachel Jose for developing the comprehension questions used with the stories and assisting with coding the data. The authors would also like to thank Dr. Eva Poort for providing feedback on an earlier version of this article.

Additional Information and Declarations

Competing Interests

Author Contributions

Human Ethics

Data Availability

1 While each new meaning comprised three distinguishing semantic features, due to the natural integration of the new word meanings into the stories not all of the semantic features were mentioned at each occurrence. Because it was more difficult to control even exposure to all of the semantic features in the incidental condition (stories), we were concerned that a scaled accuracy measure of the number of semantic features recalled would be biased towards the intentional condition. We therefore decided to score recall responses with a binary accuracy score.

2 Random slopes for the controlled factor of learning order were not included due to issues of model non-convergence. This factor was not of theoretical interest and the fixed effect for this factor was not significant in any of the models.

The authors declare that they have no competing interests.

Rachael C. Hulme conceived and designed the experiments, performed the experiments, analyzed the data, prepared figures and/or tables, authored or reviewed drafts of the paper, and approved the final draft.

Jennifer M. Rodd conceived and designed the experiments, authored or reviewed drafts of the paper, and approved the final draft.

The following information was supplied relating to ethical approvals (i.e., approving body and any reference numbers):

The UCL Experimental Psychology Ethics Committee granted ethical approval for the research (Ref: EP/2017/009).

The following information was supplied regarding data availability:

Data, analysis scripts, and materials for Experiments 1-3, as well as pre-registrations for Experiments 2-3 are available at OSF.

Experiment 1: Hulme, Rachael C, and Jennifer M Rodd. 2019. “Incidental and Intentional Vocabulary Learning for Familiar Words: The Benefit of Immediate Testing [Experiment 1].” OSF. February 27. DOI 10.17605/OSF.IO/K32TW.

Experiment 2: Hulme, Rachael C, and Jennifer M Rodd. 2021. “Incidental and Intentional Vocabulary Learning for Familiar Words: The Benefit of Immediate Testing [Experiment 2].” OSF. March 3. DOI 10.17605/OSF.IO/UPMNR.

Experiment 3: Hulme, Rachael C, and Jennifer M Rodd. 2021. “Incidental and Intentional Vocabulary Learning for Familiar Words: The Benefit of Immediate Testing [Experiment 3].” OSF. March 3. DOI 10.17605/OSF.IO/EH2C6.

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
