# Peer review of "Learning new word meanings from story reading: the benefit of immediate testing"

_PeerJ, doi:10.7717/peerj.11693_

## Round 0.1 · original submission · Minor Revisions

I have received thoughtful reviews from three experts in the field. Reviewer 1 was Lisa Henderson and Reviewer 3 was Gesa van den Broek. All reviewers are incredibly positive about the manuscript, as am I. The SoTL literature is often given a black eye for lacking control. However, the criticism fails to acknowledge that we will only gain a god’s eye view by sampling behavior across a continuum of natural (low control) and sterile (high control) settings. This work presents highly controlled experiments that can speak directly to less-controlled educational settings and thereby provides a nice SoTL bridge.

As you will see below, most of the critiques simply ask for additional justifications and clarifications throughout the document. I see none of these critiques as substantial barriers to publication, and I suspect most of them can be addressed in short order.

Prof. Henderson takes issue with your use of the word “explore” when discussing pre-registered analyses. I do not have any deep concerns about your word usage, here, and consequently leave the decision of whether to adjust your verbiage up to you.

One important concern that is echoed by all of the reviewers regards your sample size. I want to echo their sentiment that you should provide some justification for how your sample size(s) were established. Generally, I request that you add a statement to the paper confirming whether, for all experiments, you have reported all measures, conditions, data exclusions, and how you determined your sample sizes. You should, of course, add any additional text to ensure the statement is accurate. This is the standard reviewer disclosure request endorsed by the Center for Open Science [see http://osf.io/project/hadz3]. I include it in every review. Although preregistration reduces the need for some of this content, I don’t think a little bit of redundancy hurts.

Thank you for submitting your work to PeerJ. It was a pleasure to read, and I look forward to the revision. On round two, I am optimistic that I will be able to make a decision without seeking guidance from our reviewers, who have been generous with their time.

·

Basic reporting

This is a really neat, comprehensive and well-designed set of experiments that set out to address a current and important questions with clear (and broad) theoretical and practical value. Specifically, the authors examine adult participants’ ability to learn new words from stories than compared to an intention training paradigm. There were two key aims: (1) to compare performance in incidental and intentional learning paradigms, and (2) to examine the role of testing on longer-term retention. Adults learned new word meanings for familiar words better under intentional than incidental learning conditions, but there was also evidence in Exp 1 (and a trend in Exp 2) showing less forgetting for items learned incidentally (attributed to richer contexts supporting long term retention). There was very clear evidence of a testing effect in both intentional and incidental learning (and for both recall and meaning matching tests).
I have a few points about reporting of results in the Abstract:
(i)Regarding the description of results for Exp 1… “Results showed that intentional learning was more efficient”… what does “more efficient” mean and was this the case for both tasks? (ii)The abstract doesn’t connect Exp 1 and Exp 2/3. It would be useful to hint at why it’s interesting to remove the immediate test, particularly in the context of incidental learning (given testing takes it away from being so incidental). (iii) “Both cued recall and multiple-choice tests enhanced retention individually” It’s not clear what was done here. I.e., whether you examined whether immediate test performance contributes independent variance to test 2 performance, or whether you manipulated the presence of immediate recall versus immediate MC to show that both influenced test 2 as compared to having no immediate test. (iv)It would be useful to point out that the participants were adults in the abstract (given there’s a large associated literature on children).
Intro
Pg 7, lines 40-41. It’s highlighted here that incidental learning from natural linguistic environments (e.g., conversations, books, TV) is the main source of vocabulary learning, but references are taken from the developmental literature. These references might be more appropriate for supporting the idea that older children and adults learn most of their words through exposure to written texts: Nagy W., Herman P., & Anderson R. C. (1985). Learning words from context. Reading Research Quarterly, 20(2), 233–253 and Sternberg R. J. (1987). Most vocabulary is learned from context. In McKeown M. G., & Curtis M. E. (Eds.), The nature of vocabulary acquisition (pp. 89–106). Hillsdale, NJ: Erlbaum. You also cite the Batterink and Neville (2011) paper later on, which might be better placed here.
Pg 7. I find “explore” a strange choice of wording here “We explore the extent to which people’s ability to retain newly-learned word meanings over time is improved by requiring them to retrieve these word meanings during the intervening period between encoding and a later test.” given this is the pre-registered component. Perhaps it would be helpful to first outline that you’ll explore differences between incidental and intentional learning, before going on to examine the influence of testing on longer-term retention in these training paradigms.
Line 86/7. Further discussion of why it’s of interest to compare incidental and intentional learning would be helpful here. You get to this in the paragraphs starting on line 136 (intro to Exp 1) but I think it would help to lay some of this out at this point. Related to the differences between intentional/incidental instruction, you briefly mention that intentional learning might allow for more strategic processing than incidental learning. Perhaps strategic processing might be particularly important for facilitating access to prior knowledge in the case of synonym learning?
Lines 99-101. “If the presence of a quick, immediate vocabulary test can indeed enhance learning/retention for incidentally learned vocabulary this could potentially provide a simple method for boosting vocabulary gains from story reading” I can see how this could be useful in an educational setting, but not so much from recreational book reading?
When describing the caveats of the Goossens et al (2014) study, it might be useful to point out that this was a study of children and the results might differ for adults (who might be better equipped to learn more successfully from the richer contexts that stories provide, for example).
Line 128 – You say here that Exp 1 provides “a foundation for the subsequent preregistered experiments to explore learning in more detail” – what do you mean by “more detail”. There’s still a bit of a disconnect between Exp 1 and Exp 2/3 at this point for me. At this point in the paper I’m wondering whether your logic might have been that the inclusion of an immediate test might be a confounding factor in previous incidental studies, and encourage participants to deploy strategies after the story reading phase (making the whole thing less incidental)? Setting up your lines of logic in the introduction would be helpful for the reader to understand the connections between the present set of experiments.
Line 148 – what do you mean by an “efficiency advantage” for incidental learning?
In setting up the hypotheses for Exp 1, you’re not clear on whether these predictions refer to both test points. I’m presuming this was on purpose given the more exploratory nature of this experiment, but perhaps it’s worth noting that?

Experimental design

There’s no mention of how you landed at the sample size of 40 for Exp 1? Is this sample size comparable to Hulme et al (2019)? It would be useful to note whether it is equivalently powered at least.
Why did you use a children’s author? Were these stimuli designed for children?
It would be useful to clarify whether an inference was always required to extract the meaning (as in the “foam” example provided, thus enhancing the incidental nature of meaning comprehension.
Were participants permitted to re-read each page as many times as they liked? Also, you say that positioning of the words in the story was naturally distributed, but it would be useful to know how they were distributed over the pages (i.e., is it possible that all occurrences of the word to-be-learned appeared on the same page?).
Although it’s stated that the questions did not tap into the new word meanings, do you think it’s still possible that these questions could have introduced strategies that focussed towards comprehension/understanding of the new word meanings? At least, inserting questions during the comprehension process might have worked to improve comprehension, increasing metacognitive awareness of the comprehension process, so it could arguably enhance the benefits of intentional learning. This is referred to as “guided reading” in the education world. There’s some discussion of it in Blything, Hardie and Cain (2020 - https://ila.onlinelibrary.wiley.com/doi/full/10.1002/rrq.279). Perhaps worthy of a brief sentence in the Discussion as something to note for future experiment design.
As an aside, I would love to know whether participants’ vocabulary knowledge on the Mill Hill VT predicts their ability to learn (and particularly retain) words in the incidental condition more than in the intentional condition!! We’ve seen that vocabulary predicts children’s and adults’ ability to consolidate new words, but only in some experiments – it seems to be stronger when ppts are learning from richer contexts, which of course makes sense.
I’m not sure it’s described anywhere why you chose to use both the recall and meaning matching tasks, and whether you expect them to behave in the same way.
Line 313-315 – Given you assigned each new word meaning three distinctive semantic features I was surprised to read that you scored the recall responses as 0 or 1. Why not use a 0-3 scale for each item, so that depth of semantic knowledge could be captured? This would seem to me to be a more sensitive measure of semantic learning than a binary score.
Line 462 “performance will be compared to a conventional explicit learning baseline” – this is a bit ambiguous as to whether the intentional training condition is the same as that used in Experiment 1.
As in Exp 1, there’s no justification of the sample size. It would also be useful to clarify why more participants were used in Exp 2 than in Exp 1.
Line 533 – it would be useful to remind us how many participants were therefore included in the analysis.
Line 545. Apologies if I am missing something obvious, but I’m confused about the meaning matching task issue (i.e., why you used a four choice task for the immediate test but an eight choice task for the delayed test). Was this because the immediate test was administered immediately after the story/intentional training, whereas all items were put together at the delayed test? Why was this not a problem for Exp 1?

Validity of the findings

Whilst I agree with your interpretation of the test by condition interaction for meaning matching (Exp 1), it might also be worth considering the idea that memory reactivations during sleep have been to be preferentialised towards initially weak traces and less supportive to memories that already possess strong and enduring representations. One study, for example, showed a memory benefit of sleep for word pairs learned to a criterion of 60% accuracy but not for word pairs learned to a criterion of 90% accuracy (Drosopoulos, Schulze, Fischer & Born, 2007, JEPG). Thus, this could offer an important alternative explanation for your results here.
Line 731 – This could do with some rewording to tone it down slightly given the non-significant effects… “There was non-significantly less forgetting of items trained incidentally through the stories, and the testing effect was also non-significantly larger for incidentally-trained items, which seemed to benefit from the additional learning opportunity afforded by the immediate test.” (e.g., I think it would be less clumsy to talk about trends in the data that are deserving of further investigation). Also, are these non-significant interactions sufficiently powered?
1009 – 1011 – I agree that the reduced forgetting following incidental learning is really interesting and I am quite convinced by it, but I think you need to tone down the claim a little and note the need for replication given the non-significant interaction in Exp 2. Also, on line 1003 it would be useful to clarify whether the trend for reduced forgetting following incidental learning was present regardless of an immediate test, otherwise it could be the immediate test that’s contributing to the resilience from forgetting seen in Exp 1.
1041 – I *think* we are usually careful to say that overnight changes in memory in 0-24hr test studies could be consequence of both consolidation during sleep and repeat testing. There is, of course, evidence for a benefit of sleep beyond repeat testing in studies employing a 12-12 design (e.g., Henderson et al., 2012; James et al., 2020), and in Henderson et al (2013) where we omitted the immediate test for a subset of the participants and found no difference in retention at the 24 hour test (but this was admittedly in children who may show bigger effects of sleep-based consolidation). We’ve also used train-twice test once designs (e.g., Weighall et al., 2017) to reduce the effects of testing. So, whilst, yes, I agree that “some studies considering the impact of factors such as the importance of sleep for consolidation have somewhat neglected this important aspect”, I would strongly emphasize the “somewhat” and that we have also taken significant steps to address the testing effect in this work. We are also actually examining the testing effect in context of consolidation so hopefully more papers on come on this important topic!

Additional comments

This paper is a thoroughly enjoyable read. I can see it being highly cited and of use to a breadth of researchers. Really looking forward to seeing how this work progresses!

Reviewer 2 ·

Basic reporting

The three experiments are clearly reported, and the introduction makes clear how they fill an important knowledge gap in the literature. The authors have made all data and analysis scripts available; I tested a number of the analyses and was able to run the code without problems.

1) My main suggestion here is that the introduction would benefit from slightly more coverage of the possible mechanisms for retrieval practice in supporting new memories. In lines 93-108, retrieval practice is introduced as an opportunity for further learning, which to me implies re-exposure to the items rather than the possible memory mechanisms that might be involved in retention. I was pleased to see the authors cover this literature in further detail in the General Discussion (e.g., Antony et al. fast route to consolidation), but at least a brief mention of this literature in the introduction would be helpful in guiding the reader to think about the benefits of retrieval practice beyond it being an additional form of encoding/training.

2) The results section of Experiment 2 in particular is quite dense, with lots of separate follow-up comparisons for the cued recall task which make it hard to follow the key findings. It would really help to see these different simple effects and pairwise comparisons presented in a table together to clarify what each is contributing to our understanding.

3) Minor comments:
• Line 152-154 – the initial hypotheses were not entirely clear to me here, the two parts of sentence seem to contradict each other. I think the “comparably” is perhaps intended to be relative to the findings of Hulme et al. (2019), but it reads to me as if it should be comparably good with intentional learning.
• Line 462 – introducing a test of participants’ knowledge immediately after exposure? (not test?)
• Figure 4 - It may be an issue with the upload/download, but the shading in the Immediate test condition is not as clearly distinguished as it is for all other conditions/figures.

Experimental design

The research questions addressed in this paper are very clearly defined from the outset, and the authors situate the studies very clearly in the existing literature. The second two experiments were pre-registered and the authors have reported any deviations from original plans in a clear and transparent manner. The materials are available and have direct links in the text which make it very easy to view them alongside reading the manuscript. I also particularly appreciated the clear and detailed section of their modelling approach, and think it would have been possible to reproduce the analyses even without the scripts available.

I note only minor suggestions here that would help to guide the reader through changes across experiments, and a few details that would help in ensuring the methods are replicable:
• The participant section for each experiment would benefit from a sample size justification (even if this was a practical one), particularly as this changes after Experiment 1.
• Similarly, it would help to make explicit why the inclusion/analysis approach changed from Experiment 1 to 2 (i.e., not including participants who did not complete the follow-up session)
• Did the authors check for overlap in items between the trained words and filler vocabulary task?
• What was the overall premise of the study given to participants? The authors mention that they were not told of a memory test, but I am unclear over how they were introduced to the experiment.
• What were the timing restrictions on completion of the Day 2 session? It is mentioned that participants were excluded for completing the task after the deadline (e.g., line 170) and the range of completion times (line 303), but it is not clear what the acceptable time period was.

Validity of the findings

A strength of this paper is that the three experiments incorporate a number of internal replications that support the authors’ conclusions (e.g., benefit of intentional learning over incidental learning; more forgetting following intentional learning). Sometimes the findings are not quite clear cut, but the authors have been cautious in stating where effects were not statistically significant. As already mentioned in previous sections, the raw data and analysis scripts are available and easy to reproduce.

I am left with two questions over the validity of the conclusions from the present data, and a further two theoretical considerations in the General Comments Section.

1) The authors find a bigger drop in performance for items learned in the intentional condition versus the incidental condition when tested the next day. This is an interesting result that is not necessarily intuitive, and the authors suggest some possible explanations regarding the richer semantic context (lines 444, 1007). However, it could also be the case that this difference is a function of initial learning level: initial performance was very high in the intentional learning condition and had much further to fall.

2) To what extent are the authors confident that participants did not treat the incidental task as an intentional learning task, despite no instruction to do so? Adult participants in psychology experiments are often (irritatingly!) aware of task demands and what might be expected of them, and presenting known words with entirely different meanings is perhaps still very salient to a participant who is second-guessing the tasks ahead. Speaking to this, I was surprised that there was no effect of condition order, but this could suggest that participants were suspicious of the learning task in both conditions. Did the authors collect any participant feedback data that might speak to this concern?

Additional comments

I thoroughly enjoyed reviewing this manuscript: the experiments are well-designed, tackle an important gap in the literature, and I appreciated the authors’ transparency in using pre-registration and sharing all materials, data, and analysis scripts.

I have two further theoretical questions that the authors might consider addressing in their discussion:

1) To what extent could the current findings of a testing benefit be a bias rather than a concrete benefit for the items tested (i.e., also driven by the cost of *not* testing half the items)? If memory capacity is limited, then there may not be any greater improvement seen for providing an immediate test for all items. By testing each participant on only half of the items in the immediate test, those items could be more salient in memory and out-compete items not tested and regarded as less important (e.g., van Dongen et al., 2012). In contrast, a between-subjects design could show that the practice of testing actually enhances performance relative to not testing - rather than just biasing which items are remembered. Confirming that this is the case would be an important pre-requisite to determining whether test-enhanced learning provides an absolute benefit in an applied setting. I am not suggesting another experiment here, but wonder whether this is something the authors have considered or whether anything in their data/literature review speak to this question?

2) The authors make an interesting and important point regarding the use of repeat testing in studies of offline consolidation (lines 1037-1053) and, as they note in the previous paragraph, it is not clear the extent to which the testing benefit observed in these experiments relates to consolidation processes. In this regard, it would be more balanced to also note that there are many studies that have observed an offline consolidation benefit above and beyond repeat testing (e.g., those that use an AM-PM design to compare 12-hour periods associated with wake and sleep; and those that also include a further 24-hour test). The present study is valuable in highlighting the importance of considering *both* aspects together, especially following incidental learning contexts which I think has been somewhat neglected, but does not exclude a role for sleep-based mechanisms in previous (or the present) experiments.

·

Basic reporting

The reporting is clear, unambiguous and complete.

There are two aspects that I think need to be worked out more clearly in the theoretical introduction to provide context.

One is that the distinction between incidental and intentional learning seems to be mixed with the distinction of contextualized and non-contextualized learning at times. Perhaps incidental learning is usually contextualized, but intentional learning can also involve context or inferences of word meaning from context (as a case in point, the authors refer to a publication in which we showed benefits of retrieval practice when participants intentionally practiced vocabulary in a sentence context).

Second, and perhaps related, I miss an argument why the testing effect might be different depending on the type of encoding that preceded retrieval practice. The authors give a relatively broad explanation, suggesting that incidental and intentional encoding might trigger different kinds of information processing (one involving more semantic elaboration, one leading to more attention to the word form, ll. 136ff) and that “it is unclear whether the testing effect would provide a similar benefit for vocabulary learned under incidental learning conditions” (ll. 95f). Yet, it remains open how or why these differences in encoding would moderate the testing effect. I think by providing information on relevant cognitive mechanisms of retrieval practice, this question could be introduced in a more informative and more convincing way. Two suggestions that might help build an argument why incidental versus intentional encoding could influence subsequent retrieval practice differently: (1) there is a debate in the literature on retrieval practice regarding the importance of semantic elaboration and/or reinstatement of the encoding context during retrieval (most cited publications are from the labs of Carpenter or Karpicke, for example). You could possibly link these to your argument that the incidental learning condition may have involved more semantic elaboration or a richer encoding context. (2) the meta-analysis by Rowland (2014) and other publications suggest that benefits of retrieval practice are larger when retrieval success during practice is high, and retrieval success may depend on (the success of) prior encoding. Thus, if intentional encoding is more effective than incidental encoding, that could cause the testing effect to be larger after intentional encoding.

Textual suggestions
l. 151: Quote: “Based on the previous research, we predicted that learning of new word meanings would be better for intentional learning conditions, although we expected comparably good vocabulary learning for the incidental learning conditions in line with the findings of Hulme et al. (2019).”
I may misread the sentence, but the two subclauses seem to contradict one another.

ll. 935. “Although accuracy…” maybe double-check this sentence. It refers to an interaction effect but (seems to) describes a main effect.
l.374, should “although” be “however”?
l.461 remove “a”
l.743 “test” should be “testing”
l. 907 “is” should be “it”
l. 1044 Check use of “Although”

Experimental design

This review follows the structure of the three-experiment paper and not all comments clearly fit into one of the three sections, therefore, the majority is posted here and the comments on conclusions and wider interpretation of findings in the general discussion are included under "Validity of findings".

General comment: Enough information is provided on the methodology of the experiments, which appears sound. The statistical models are described clearly and systematically so that, in spite of the large number of models, it is easy to follow how different effects were tested and how models were selected, including the simple contrast analyses done to follow up on interaction effects. Authors also clearly distinguished between preregistered analyses and analyses that deviated from the preregistered analysis plan.

Experiment 1
Results
- Figure 1: If I understand correctly, the mean percentage accuracy was calculated per participant across a number of trials. Could you include information about the distribution of values of participants in the plot, for example, by showing a violin plot instead of a bargraph, or by including the raw data observations (for example, using the jitter function in R)? I know that this is a remark that is easily written in a review yet costs the authors plenty of time but if I understand correctly what is depicted, then this adjustment would make the plots more informative.
- There is an order by condition interaction for the data of Day 1 (ll. 374ff). The authors write “This result was most likely driven by participants becoming somewhat fatigued by the second task and this fatigue effect having greater impact on the story reading task, which took longer.”. I don’t find this a convincing explanation because performance was actually (numerically) higher for the intentional condition when that condition was the second task rather than the first task; thus, there seems to be no general fatigue effect, and in particular no such effect after the longer story reading task.

Experiment 2
Methods
- It speaks for this study that the authors recruited sufficient participants to work with an unannounced second test. Based on the participant section, I thought that the data of 99 participants were analysed but after reading the procedure (l.533), I got doubts whether you analysed data of 99-36=63 participants or 99 participants. Perhaps these sections can be edited to avoid confusion.

Experiment 3
Introduction
A previous study that contrasted cued recall and multiple choice practice tests in vocabulary learning, including interactions of practice test and posttest format, is Nakata 2016 (doi: 10.1515/iral-2015-0022). I think this should be referenced here or in the general discussion.

Validity of the findings

General Discussion
While reading, I found the three experiments only loosely connected (proceeding from Experiment 1 on incidental versus intentional learning to Experiment 2 on testing effects to Experiment 3 on test format effects). The general discussion brought Experiment 1 and 2 together a bit more, but there is no explicit discussion of Experiment 3. Why is that?

I miss a discussion of the specific characteristics of the stories used in this experiment and how these may have influenced how effectively participants learned the new word meanings in the story condition (e.g., that/whether the target words were central to the story and repeated multiple times, whether the stories were interesting/appealing for the participant audience). In turn, (how) does this limit the generalizability of findings?

Speculation: Regarding the interpretation of enhanced forgetting over time in the intentional condition, an alternative interpretation that could be valuable to add is that the intentional condition brought some weakly encoded items just above the recall threshold for the immediate test, while the incidental condition did not (thus resulting in better immediate test results). If these weaker items fell below the recall threshold of the delayed posttest and thus delayed posttest scores were lower than immediate scores for the intentional encoding condition, is it then fair to say that items were more susceptible to forgetting in the intentional condition compared to the incidental condition? (for a discussion of how distributions of item strengths could cause patterns of posttest findings, in the retrieval practice literature, see Kornell et al, 2011, doi:10.1016/j.jml.2011.04.002 ).

Minor: The explanation of benefits of intentional learning as the intentional task involving some kind of multiple choice retrieval practice (ll.993ff), would be interesting to connect to the finding of Experiment 2 that adding the immediate test had a stronger benefit (testing effect) on incidental learning than intentional learning (possibly because intentional learning already involved a limited testing effect?).

Additional comments

I appreciated the clear writing of this paper and the logical structure, and I applaud the authors’ choice for preregistration and sharing their data and materials online in a format that can easily be reviewed. Before receiving this review invitation I had actually seen the materials of the project online, so they do find their way into the research community and I hope that the open access will lead to their use in future research. The stories are well-written and an interesting tool to study incidental word learning.

I recommend a revision before publication of the paper and I consider most of my suggestions minor revisions.

---

## Round 0.2 · accepted · Accept

I did not send your manuscript out for a second round of review. I believe you have adequately addressed the reviewers' critiques (which were minimal to begin with). Congratulations. This is a wonderful contribution to the literature.